# A Cramér–von Mises Approach to Incentivizing Truthful Data Sharing

**Alex Clinton**
University of Wisconsin-Madison
aclinton@wisc.edu

**Thomas Zeng**
University of Wisconsin-Madison
tpzeng@wisc.edu

**Yiding Chen**
Cornell University
yc2773@cornell.edu

**Xiaojin Zhu**
University of Wisconsin-Madison
jerryzhu@cs.wisc.edu

**Kirthevasan Kandasamy**
University of Wisconsin-Madison
kandasamy@cs.wisc.edu

## Abstract

Modern data marketplaces and data sharing consortia increasingly rely on incentive mechanisms to encourage agents to contribute data. However, schemes that reward agents based on the quantity of submitted data are vulnerable to manipulation, as agents may submit fabricated or low-quality data to inflate their rewards. Prior work has proposed comparing each agent's data against others' to promote honesty: when others contribute genuine data, the best way to minimize discrepancy is to do the same. Yet prior implementations of this idea rely on very strong assumptions about the data distribution (e.g. Gaussian), limiting their applicability. In this work, we develop reward mechanisms based on a novel two-sample test statistic inspired by the Cramér-von Mises statistic. Our methods strictly incentivize agents to submit more genuine data, while disincentivizing data fabrication and other types of untruthful reporting. We establish that truthful reporting constitutes a (possibly approximate) Nash equilibrium in both Bayesian and prior-agnostic settings. We theoretically instantiate our method in three canonical data sharing problems and show that it relaxes key assumptions made by prior work. Empirically, we demonstrate that our mechanism incentivizes truthful data sharing via simulations and on real-world language and image data.

## 1 Introduction

Data is invaluable for machine learning (ML). Yet many organizations and individuals lack the capability to collect sufficient data on their own. This has driven the emergence of *data marketplaces* [1–3]—where consumers purchase data from contributors with money—and *consortia* [4–6] for data sharing and federated learning—where agents share their own data in return for access to others' data. As such platforms depend critically on data from contributing agents, they incentivize these agents to contribute more data via commensurate rewards: consortia typically grant agents greater access to the pooled data [7, 8], while marketplaces provide correspondingly larger payments [9, 10].

However, most existing work implicitly assume that contributors will report data truthfully. In reality, strategic contributors may untruthfully report data to exploit the incentive scheme. As one such example, they may *fabricate data*—either through naïve random generation or sophisticated ML-based synthesis —to artificially inflate their submissions and maximize their own rewards. In naive incentive schemes, where rewards scale with the *quantity* of data, such behavior can flood the system with poor quality data which undermines trust in the platform.

The central challenge in preventing such strategic misreporting, including fabrication, is that consortia and marketplace operators typically lack ground-truth knowledge about the underlying data

distribution—if the ground truth were known, the very need for learning and data sharing would be obviated. To address this, prior work has proposed a simple and intuitive idea: compare each agent's data submission against the pooled submissions of other agents. In these mechanisms, when all agents' data come from the same distribution, truthful reporting constitutes a Nash equilibrium. Intuitively, when others contribute genuine data, minimizing the discrepancy between one's own submission and the aggregate submission of others also requires submitting genuine data.

Despite this promising intuition, prior work has succeeded only under strong assumptions about data distributions [7, 11] and/or narrow models of untruthful behavior [12–14]. Realizing this idea to general data distributions and arbitrary types of strategic misreporting has remained challenging.

**Our contributions.** This gap motivates the central premise of our work. We develop a mechanism where agents are rewarded based on a novel loss function that is inspired by two-sample testing. Our loss function, resembling the Cramér-von Mises (CvM) two-sample test statistic [15, 16], is computationally inexpensive, and applies to many different data types, including complex data modalities such as text and images. We design (approximate) Nash equilibria in which agents are incentivized to truthfully report data, without relying on restrictive assumptions about the underlying distribution or strategic behaviors. We theoretically demonstrate the application of our mechanism in three data sharing problems involving purchasing data, and data sharing without money. We empirically demonstrate its usefulness via experiments on synthetic and real world datasets.

## 1.1 Overview of Contributions

**Model.** There are $m$ agents. Each agent $i$ possesses a dataset $X_i$ drawn from an unknown distribution $\mathcal{P}$, and submits $Y_i$, not necessarily truthfully (i.e. $Y_i \neq X_i$). In data-sharing consortia or marketplaces, the goal is to design losses (negative rewards) $L = \{L_i\}_{i \in [m]}$, where agent $i$ is rewarded according to $-L_i(\{Y_j\}_{j \in [m]})$, so as to incentivize truthful reporting. A natural and widely adopted approach [7, 11, 10], which we also follow, is to design $L_i$ as a function of the form $L_i(Y_i, Y_{-i})$, where $Y_{-i} = (Y_j)_{j \neq i}$ is the pooled submission of all agents except $i$. A high value of $L_i(Y_i, Y_{-i})$ suggests that agent $i$'s data deviates from the rest, which may indicate untruthful behavior when other agents report truthfully (i.e. $Y_j = X_j$ for all $j \neq i$).

Comparing an agent's submission to the pooled data from others can be naturally viewed as computing a *two-sample test statistic*—or simply, a *two-sample test*—between $Y_i$ and $Y_{-i}$ [15, 17]. This perspective motivates the design of our loss function.

**Key technical challenges.** There are two primary challenges in designing a loss. First, we should ensure that the loss $L$ is *truthful*: specifically, when $Y_{-i}$ is drawn i.i.d. from $\mathcal{P}$ (i.e. all other agents report truthfully), the optimal strategy for agent $i$ to minimize $L_i(Y_i, Y_{-i})$ should be to also submit truthfully, i.e. $Y_i = X_i$. Without this property, agents may have an incentive to manipulate their submissions to reduce $L_i(Y_i, Y_{-i})$. However, many standard two-sample tests—such as Kolmogorov–Smirnov [17, 18], $t$-test [19], Mann–Whitney [20], and MMD [21]—are not provably truthful. The second challenge is to reward agents for higher quality submissions, i.e. $L_i$ should decrease as the quantity of the submitted (truthful) data increases.

While each challenge is easy to address in isolation, satisfying both simultaneously is far more difficult. For example, a mechanism that rewards agents equally is trivially truthful but offers no incentive to collect more data. Conversely, if losses are tied solely to the quantity of submitted data, the mechanism becomes vulnerable to data fabrication, leaving honest agents worse off.

A third, less central challenge is ensuring that we have a handle on the distribution of $L_i$ to enable its application in data sharing use cases. For instance, penalizing large values of $L_i$ requires understanding what constitutes "large" under truthful reporting. Prior work addresses these three challenges only under strong assumptions on $\mathcal{P}$ (e.g. Gaussian [7, 11], Bernoulli [22], restricted class of exponential families [10]), or narrow models of untruthful reporting [12, 13, 22].

**Our method and results.** In §2, we consider a Bayesian setting in which each agent's data is drawn from an *unknown* distribution $\mathcal{P}$, itself sampled from a *known* prior $\Pi$. We introduce our loss $L$ which is inspired by the Cramér–von Mises (CvM) test. Leveraging this statistic along with user-specified data featurizations, we design a loss in which truthful reporting forms an exact Nash equilibrium (NE). Moreover, we show that $L$ incentivizes the submission of larger datasets—an agent is *strictly* better off by submitting more truthful data. Our loss is also bounded, and decreases gracefully with the amount of data submitted, making it useful for data sharing applications as we will see in §4.

However, this approach has two practical limitations. First, specifying a meaningful prior can be difficult, particularly for complex data modalities such as text or images. Second, even with a prior, computing $L$ may be intractable when it requires expensive Bayesian posterior computations. In §3, we address these issues by replacing the above Bayesian version of our loss with a prior-agnostic version that is simpler to compute. We show that this leads to a truthful $\varepsilon$-approximate NE in both Bayesian and frequentist settings where $\varepsilon$ approaches zero as the amount of data submitted increases. We also show that agents benefit from submitting more data, and that our new loss is also bounded and decreases gracefully with the amount of data submitted.

**Applications.** In §4, we theoretically demonstrate how our Bayesian method can be applied to solve three different data sharing problems, some of which have been studied in prior work, while relaxing their technical conditions. The first problem is incentivizing truthful data submissions via payments assuming agents already possess data [10]. The second is the design of a data marketplace where a buyer is willing to pay strategic agents to collect data on her behalf [23]. The third is a federated learning setting where agents wish to share data for ML tasks without the use of money [8].

**Empirical evaluation.** In §5, we empirically evaluate our methods on simulations, and real world image and language experiments. To simulate untruthful behavior, we consider agents who augment their datasets by fabricating samples using simple fitted models, or generative models such as diffusion models and LLMs [24–26]. Our results demonstrate that such untruthful submissions lead to larger losses compared to truthful reporting. This corroborates theoretical results for both methods and demonstrates that the prior-agnostic version is practically useful for real world data sharing.

## 1.2 Related Work

There has been growing interest in the incentive structures underlying data sharing, federated learning, and data marketplaces. A central goal in these settings is to incentivize data contributions. However, most prior work do not consider untruthful reporting. When they do, they either impose restrictive distributional assumptions, or limit how contributors may misreport.

**Incentivizing data sharing without truthfulness requirements.** A line of work addresses incentivizing data collection in federated learning [27, 8, 28–31, 9]. Other studies focus on incentivizing the sharing of private data [32] or truthful reporting of private data collection costs [14]. All of these works assume agents report data truthfully, and do not encounter the challenges we address here.

**Restricted distributional assumptions.** Cai et al. [9] study a principal-agent model where a principal selects measurement locations and compensates agents who exert costly effort to reduce observation noise. Their optimal contract relies on a known effort-to-data-quality function, which may be unknown or nonexistent in practice. Ghosh et al. [22] design a mechanism to purchase binary data under differential privacy, compensating agents for privacy loss. Chen et al. [10] drop the privacy constraint to handle non-binary data, proposing a fixed-budget mechanism that ensures truthful reporting, but requiring the data distribution to have finite support or belong to an exponential family. Other work focuses on incentivizing truthful reporting in Gaussian mean estimation for data sharing [7, 11] and data marketplaces [23]; however, as our experiments show, their approach—based on comparing means of the reported data—does not generalize beyond Gaussian data.

**Restricted untruthful reporting.** Falconer et al. [13] propose monetary incentives for data sharing, assuming agents can only fabricate data by duplicating existing entries. Dorner et al. [12] study mean estimation where agents may misreport only by adding a scalar to their true values.

**Peer prediction.** The peer prediction literature addresses a challenge similar to ours: eliciting truthful reports without access to ground truth. Prior work [33–36] uses reported signals to cross-validate agents' submissions, showing that truthful reporting forms an (approximate) Nash equilibrium. Techniques from [37, 38] have been applied to design payment-based mechanisms for data sharing [10], but these rely on strong assumptions about the data distribution (e.g., exponential families or finite support). It is not clear if these methods generally work when agents may change the number of signals (data points) they have, which is a critical consideration in data sharing use cases where fabrication is possible. More precisely, the mechanism designer does not know how many data points an agent holds, yet must still incentivize truthful reporting.

*Practical applicability.* The vast majority of the above works focus on theoretical development, but lack empirical evaluation, with their practicality unclear due to expensive Bayesian computations. In contrast, our prior-agnostic method is simple and performs well on real data.

**Review of the Cramér-von Mises test.** We briefly review the Cramér–von Mises (CvM) test [15]. Let $X = \{X_1, \ldots, X_n\} \overset{\text{i.i.d.}}{\sim} F_1$ and $Y = \{Y_1, \ldots, Y_m\} \overset{\text{i.i.d.}}{\sim} F_2$ be samples from $\mathbb{R}$-valued distributions $F_1$ and $F_2$, respectively. Let $F_X(t) = \frac{1}{|X|} \sum_{x \in X} 1_{\{x \leq t\}}$ and $F_Y(t) = \frac{1}{|Y|} \sum_{y \in Y} 1_{\{y \leq t\}}$ be the empirical CDFs (ECDFs) of $X$ and $Y$. Set $Z = (X_1, \ldots, X_n, Y_1, \ldots, Y_m)$. The two-sample CvM test statistic is then defined below in (1). We have illustrated the CvM test in Fig. 1a.

$$\text{CvM}(X, Y) = \frac{nm}{(n+m)^2} \sum_{i=1}^{n+m} (F_X(Z_i) - F_Y(Z_i))^2. \tag{1}$$

## 2 A Truthful Mechanism in a Bayesian Setting

In this section, we design a mechanism to reward agents based on the quality of their submitted data. We begin by specifying our model. To build intuition, we present a simplified single-variable version of our loss (mechanism) in §2.1. We then present the general version of our mechanism in §2.2.

**Setting.** There are $m > 2$ agents, where each agent $i \in [m]$ has a dataset $X_i = \{X_{i,1}, \ldots, X_{i,n_i}\} \subset \{X_{i,j}\}_{j=1}^{\infty}$ of $n_i \in \mathbb{N}$ points. Here $\{X_{i,j}\}_{j=1}^{\infty}$ are drawn i.i.d. from an *unknown* distribution $\mathcal{P}$ over $\mathcal{X}$ and $X_i \in \mathcal{X}^{n_i}$. We refer to $\mathcal{X}$ as the dataspace; examples include the space of images, text, or simply $\mathbb{R}^d$. In this section, we consider a Bayesian setting where $\mathcal{P}$ is drawn from a *publicly known* prior $\Pi$. A mechanism designer wishes to incentivize the agents to report their datasets truthfully by designing losses (negative rewards).

Let $\mathcal{D} = \bigsqcup_{\ell=0}^{\infty} \mathcal{X}^\ell$ be the collection of finite subsets of $\mathcal{X}$, which forms the space of datasets an agent could possess. A mechanism for this problem is a normal form game which maps the agents' dataset submissions to a vector of losses, i.e. $L \in \{L' : \mathcal{D}^m \to \mathbb{R}^m\}$. Once the mechanism $L$ is published, each agent will submit a dataset $Y_i$ (not necessarily equal to $X_i$). An agent's strategy can be viewed as a function $f_i \in \mathcal{F} = \{f : \mathcal{D} \to \mathcal{D} \text{ s.t. } f \text{ is measurable}\}$ which maps their original dataset $X_i$ to $Y_i = f_i(X_i)$. This allows for strategic data manipulations which may depend on the agent's own dataset. Let $I$ be the identity (truthful) strategy which maps a dataset to itself, i.e. $I(X_i) = X_i$.

Agent $i$'s loss $L_i$ is the $i$'th ouput of the mechanism $L$, and is a function of the strategies $f = \{f_i\}_{i \in [m]}$ adopted by other agents and the initial datasets $X = \{X_1, \ldots, X_m\}$, and can be written as $L_i = L_i(\{f_i\}_{i \in [m]}) = L_i(\{f_i(X_i)\}_{i \in [m]})$ to highlight or suppress these dependencies.

**Requirements.** The mechanism designer wishes to design $L$ to satisfy two key properties:

1. *Truthfulness:* All agents submitting truthfully ($f_i = I$), is a Nash equilibrium, that is,
$$\forall i \in [m], \forall f_i \in \mathcal{F}, \quad \mathbb{E}\left[L_i(\{I\}_{j=1}^m)\right] \leq \mathbb{E}\left[L_i(f_i, \{I\}_{j \neq i})\right].$$

2. *More (data) is (strictly) better (MIB):* Let $X_i, X_i'$ be two datasets such that $|X_i'| > |X_i|$. Then,
$$\mathbb{E}\left[L_i(I(X_i'), \{I(X_j)\}_{j \neq i})\right] < \mathbb{E}\left[L_i(\{I(X_j)\}_{j \in [m]})\right].$$

Above, the expectation is with respect to the prior $\mathcal{P} \sim \Pi$, the data $X_i, X_i' \sim \mathcal{P}$ for all $i$, and any randomness in the agent strategies $f_i$ and mechanism $L$. As discussed in §1.1 under 'Key technical challenges', while satisfying either of these requirement is easy, designing a mechanism which satisfies both simultaneously is significantly more difficult.

### 2.1 Warm-up when $\mathcal{X} = \mathbb{R}$

**Algorithm 1 description.** To build intuition, we first study the simple one-dimensional case $\mathcal{X} = \mathbb{R}$. The mechanism works by aggregating all of the submissions $\{Y_i\}_{i=1}^m$ and for each agent $i \in [m]$, computing a (randomized) loss $L_i$. To compute $L_i$, an evaluation point $T_i$ is first randomly sampled from the data submitted by the other agents $Y_{-i}$. The remaining data $Z_i$ is used to define the empirical CDF $F_{Z_i}$. The loss $L_i$ is then defined as the squared difference between this ECDF evaluated at $T_i$, i.e. $F_{Z_i}(T_i)$, and its conditional expectation given $(X_{i,1}, \ldots, X_{i,|Y_i|}, T_i)$ evaluated at $(Y_{i,1}, \ldots, Y_{i,|Y_i|}, T_i)$. Finally, the mechanism outputs $L_i \in [0, 1]$ as agent $i$'s loss.

*Design intuition:* The conditional expectation $\mathbb{E}\left[F_{Z_i}(T_i) | X_{i,1}, \ldots, X_{i,|Y_i|}, T_i\right]$ can be thought of as the best guess for $F_{Z_i}(T_i)$ having seen $(X_{i,1}, \ldots, X_{i,|Y_i|})$. Thus, $\mathbb{E}\left[F_{Z_i}(T_i) | X_{i,1} = Y_{i,1}, \ldots, X_{i,|Y_i|} = Y_{i,|Y_i|}, T_i\right]$ can be thought of as the best guess for $F_{Z_i}(T_i)$

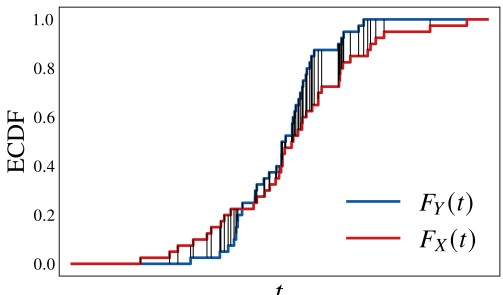
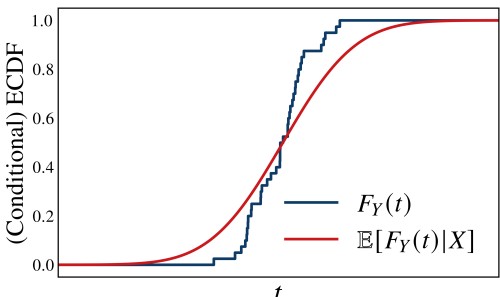

(a) The two-sample Cramér-von Mises test

(b) An empirical CDF vs its conditional expectation

Figure 1: Subfigure (a) shows the empirical CDFs (ECDF) for two datasets $X = \{X_1, \ldots, X_n\}$, $Y = \{Y_1, \ldots, Y_m\}$. The gray lines are the differences between the two curves at each point in $(X_1, \ldots, X_n, Y_1, \ldots, Y_m)$, and are used to calculate the two-sample CvM test in (1). Subfigure (b) replaces $F_Y(t)$ with $\mathbb{E}[F_Y(t)|X]$ which can be thought of as the best approximation to $F_Y(t)$ based on having seen $X$.

---

**Algorithm 1** A single variable Cramér–von Mises style statistic

---

1: **Input parameters**: A prior $\Pi$ over the set of $\mathbb{R}$-valued distributions.
2: **for** each agent $i \in [m]$:
3:      $Y_{-i} \leftarrow (Y_{j,\ell})_{j \neq i, \ell \in |Y_j|}$.
4:      Sample $j \sim \text{Unif}(1, \ldots, |Y_{-i}|)$ and set $T_i \leftarrow Y_{-i,j}, \quad Z_i \leftarrow (Y_{-i,\ell})_{\ell \neq j}$.
5:      Return $L_i \leftarrow \left( \mathbb{E}\left[ F_{Z_i}(T_i) | X_{i,1} = Y_{i,1}, \ldots, X_{i,|Y_i|} = Y_{i,|Y_i|}, T_i \right] - F_{Z_i}(T_i) \right)^2$.

---

assuming that $\left(Y_{i,1}, \ldots, Y_{i,|Y_i|}\right)$ is the agent's true data. A visual comparison of $F_{Z_i}(T_i)$ to $\mathbb{E}\left[ F_{Z_i}(T_i) | X_{i,1}, \ldots, X_{i,|Y_i|}, T_i \right]$ can be seen in Fig. 1b.

The loss $L_i$ defined above is well-posed and computable. As demonstrated in our experiments (with derivations in Appendix E), closed-form expressions for $L_i$ can be derived in simple conjugate settings such as Gaussian-Gaussian and Bernoulli-Beta, enabling efficient implementations. For more complex prior distributions, numerical approximations using methods such as MCMC [39] or variational inference [40] can be employed.

**Theoretical results.** We now present the theoretical properties of Algorithm 1. To satisfy the MIB condition, we require that the prior $\Pi$ meet a non-degeneracy condition, formalized in Definition 1. Intuitively, this condition ensures that the posterior changes upon observing an additional data point. Examples of degenerate priors include those that select a fixed distribution $\mathcal{P}$ with probability 1, or choose $\mathcal{P}$ to be a degenerate distribution $\delta_x, x \in \mathcal{X}$ with probability 1. In such cases, data sharing is meaningless, as the distribution is either fully known or revealed by a single sample. Thus, it is natural to assume $\Pi$ is non-degenerate, so that additional data remains informative.

**Definition 1.** *(Degenerate priors): Let $\mathcal{P} \sim \Pi$ and $\{X_i\}_{i=1}^{\infty}, T, Z \overset{i.i.d.}{\sim} \mathcal{P}$. We say that $\Pi$ is degenerate if for some $n \in \mathbb{N}$, $P(Z \leq T | T, X_1, \ldots, X_n) \overset{a.s.}{=} P(Z \leq T | T, X_1, \ldots, X_{n+1})$.*

Theorem 1 shows that Algorithm 1 satisfies truthfulness for all priors $\Pi$, and MIB when $\Pi$ is not degenerate. The key idea for truthfulness is that by computing the aforementioned conditional expectation, the mechanism performs, on behalf of agent $i$, the best possible guess for $F_{Z_i}(T_i)$ just using $Y_i$. Thus, it is in agent $i$'s best interest if $Y_i = X_i$.

**Theorem 1.** *The mechanism in Algorithm 1 satisfies truthfulness. Moreover, when $\Pi$ is not degenerate, then Algorithm 1 also satisfies MIB.*

While the previous theorem indicates that submitting more data is beneficial for the agent, it does not quantify how an agent's loss decreases as they contribute more data. The following proposition quantifies this by offering bounds on how an agent's expected loss decreases with the amount of data they submit, assuming all agents are truthful. This handle on $\mathbb{E}[L_i]$, along with the property that $L_i \in [0, 1]$, is useful for applying our mechanism to data sharing applications as we will see in §4.

---

**Algorithm 2** A feature-based Cramér–von Mises style statistic

---

1: **Input parameters**: A prior $\Pi$ over the set of $\mathcal{X}$-valued distributions, feature maps $\{\varphi^k\}_{k=1}^K$.
2: **for** each agent $i \in [m]$:
3:      $Y_{-i} \leftarrow (Y_{j,\ell})_{j \neq i, \ell \in |Y_j|}$.
4:      Sample $j \sim \text{Unif}(1, \ldots, |Y_{-i}|)$ and set $\;T_i \leftarrow Y_{-i,j}, \;\; Z_i \leftarrow (Y_{-i,\ell})_{\ell \neq j}$.
5:      **for** each feature $k \in [K]$:
6:          $Z_i^k \leftarrow \big(\varphi^k(Z_{i,j})\big)_{j=1}^{|Z_i|}, \;\; T_i^k \leftarrow \varphi^k(T_i)$.
7:          $L_i^k \leftarrow \Big( \mathbb{E}\Big[ F_{Z_i^k}(T_i^k) \,\big|\, X_{i,1} = Y_{i,1}, \ldots, X_{i,|Y_i|} = Y_{i,|Y_i|}, T_i^k \Big] - F_{Z_i^k}(T_i^k) \Big)^2$.
8:      Return $L_i \leftarrow \frac{1}{K} \sum_{k=1}^K L_i^k$.

---

**Proposition 1.** *Let $L_i(\{I\}_{i=1}^m)$ denote the value of $L_i$ when agents are truthful in Algorithm 1. Then,* $0 \leq \mathbb{E}[L_i(\{I\}_{i=1}^m)] \leq \frac{1}{4}\left(\frac{1}{|X_i|} + \frac{1}{|Z_i|}\right)$. *Moreover, when $\Pi$ is a prior over the set of continuous* $\mathbb{R}$*-valued distributions,* $\frac{1}{6|Z_i|} \leq \mathbb{E}[L_i(\{I\}_{i=1}^m)] \leq \frac{1}{6}\left(\frac{1}{|X_i|} + \frac{1}{|Z_i|}\right)$.

## 2.2 A General Mechanism with Feature Maps

We now extend our mechanism and to handle data from arbitrary distributions. The key modification is the introduction of feature maps: functions chosen by the mechanism designer that transform general data distributions into $\mathbb{R}$–valued distributions to apply our mechanism to.

**Feature maps.** We define a feature map to be any measurable function $\varphi : \mathcal{X} \to \mathbb{R}$ which maps the data to a single variable distribution. We will see that any collection of feature maps $\{\varphi^k : \mathcal{X} \to \mathbb{R}\}_{k=1}^K$ which map the data to a collection of single variable distributions supports a truthful mechanism. However, some feature maps perform better than others depending on the use case, so we allow the mechanism designer flexibility to select maps. For Euclidean data, coordinate projections may suffice, while for complex data like text or images, embeddings from deep learning models are more appropriate (as used in our experiments in §5).

**Algorithm 2 description.** The mechanism designer first specifies a collection of feature maps, $\{\varphi^k\}_{k=1}^K$ based on the publicly known prior $\Pi$. After this, Algorithm 2 can be viewed as applying Algorithm 1 for each feature $k \in [K]$, making use of $\varphi^k$ to map general data in $\mathcal{X}$ to $\mathbb{R}$.

The following theorem shows that Algorithm 2 is truthful, which is a result of the same arguments made in Theorem 1, now repeated for each feature map. For MIB, we require an analogous condition to the one given in Theorem 1, stating that more data leads to a more informative posterior distribution for at least one of the $K$ features. To state this formally, we first extend Definition 1.

**Definition 2.** *Let $\mathcal{P} \sim \Pi$ and $\{X_i\}_{i=1}^\infty, T, Z \overset{i.i.d.}{\sim} \mathcal{P}$. We say that $\Pi$ is degenerate for feature $k \in [K]$ if for some $n \in \mathbb{N}$,*

$$P\left(\varphi^k(Z) \leq \varphi^k(T)|\varphi^k(T), X_1, \ldots, X_n\right) \overset{a.s.}{=} P\left(\varphi^k(Z) \leq \varphi^k(T)|\varphi^k(T), X_1, \ldots, X_{n+1}\right).$$

**Theorem 2.** *The mechanism in Algorithm 2 satisfies truthfulness. Moreover, if there is a feature $k \in [K]$, for which $\Pi$ is not degenerate, then Algorithm 2 also satisfies MIB.*

Proposition 9 (Appendix C.2), analogous to Proposition 1, quantifies how $L_i$ decreases with data size, which will be useful when using this loss in data sharing applications. Additionally, Proposition 8 (Appendix C.2) gives an explicit relationship for how the expected loss changes when an agent submits an additional data point, depending on the prior and feature maps. This exactly quantifies how much lower an agent's loss is when submitting more data.

## 3 A Prior Agnostic Mechanism

While our mechanism in §2 applies broadly in Bayesian settings, it has two practical limitations. First, specifying a meaningful prior can be difficult, especially for complex data like text or images. Second, even with a suitable prior, computing the conditional expectation in line 7 may be intractable due to

---

**Algorithm 3** A prior free Cramér–von Mises style statistic

---

1: **Input parameters**: Feature maps $\{\varphi^k\}_{k=1}^K$ and an augment split map $\psi : \mathbb{N} \to \mathbb{N} : \psi(n) < n-1$.
2: **for** each agent $i \in [m]$:
3:      $Y_{-i} \leftarrow (Y_{j,\ell})_{j \neq i, \ell \in |Y_j|}$.
4:      Split $Y_{-i}$ into $Y_{-i} = (\{T_i\}, W_i, Z_i)$ s.t. $|W_i| = \psi(|Y_{-i}|)$.
5:      **for** each feature $k \in [K]$:
6:          $T_i^k \leftarrow \varphi^k(T_i)$, $\ W_i^k \leftarrow \left(\varphi^k(W_{i,\ell})\right)_{\ell=1}^{|W_i|}$, $\ Z_i^k \leftarrow \left(\varphi^k(Z_{i,\ell})\right)_{\ell=1}^{|Z_i|}$
7:          $L_i^k \leftarrow \left(F_{(Y_i^k, W_i^k)}(T_i^k) - F_{Z_i^k}(T_i^k)\right)^2$.
8:      Return $L_i \leftarrow \frac{1}{K} \sum_{k=1}^K L_i^k$.

---

the cost of Bayesian posterior inference. To address this, we introduce a prior-agnostic variant that is significantly easier to compute. The trade-off is that truthful reporting becomes an $\varepsilon$-approximate NE, where $\varepsilon$ vanishes as the amount of submitted data grows.

**Changes to Algorithm 2.** Thus far, we have only focused on the Bayesian setting, assuming that agents wish to minimize their expected loss $\mathbb{E}_{\mathcal{P} \sim \Pi}\left[\mathbb{E}_{\{X_i\}_{i=1}^m \sim \mathcal{P}}[L_i]\right]$. However, this modification also supports a frequentist view where agents wish to minimize their worst case expected loss over a class $\mathcal{C}$ possible distributions, i.e. $\sup_{\mathcal{P} \in \mathcal{C}} \mathbb{E}_{\{X_i\}_{i=1}^m \sim \mathcal{P}}[L_i]$. In the frequentist setting, the class $\mathcal{C}$ is the analog of the prior $\Pi$. As such, our prior agnostic mechanism does not have a prior $\Pi$ as input.

Algorithm 3 computes each agent's loss as follows: first partition $Y_{-i}$ into three parts, (1) an evaluation point $T_i$, (2) data to augment agent $i$'s submission with $W_i$, and (3) data to compare agent $i$'s submission against $Z_i$. The mechanism designer is free to choose how much data to allocate to $W_i$ as given by the map $\psi$. For each feature $k \in [K]$, we then obtain $T_i^k$, $W_i^k$, and $Z_i^k$ by applying $\varphi^k$. The main modification of the prior-agnostic mechanism is that the conditional expectation in line 7 of Algorithm 2, $\mathbb{E}[F_{Z_i^k}(T_i^k) \,|\, X_i = Y_i, T_i^k]$, is replaced with $F_{(Y_i^k, W_i^k)}(T_i^k)$ which serves as an easy to compute estimate for $F_{Z_i^k}(T_i^k)$. Here $F_{(Y_i^k, W_i^k)}$ denotes the ECDF from the combined data of $Y_i^k$ and $W_i^k$. The reason we allow the mechanism designer the flexibility to supplement $Y_i^k$ with $W_i^k$ is that doing so allows them to decrease the $\varepsilon$ parameter corresponding to truthfulness being an $\varepsilon$-approximate Nash in the following theorem. A reasonable choice for the size of $W_i$ is to set it so that $|W_i| + |Y_i| = |Z_i|$.

Before stating the theorem, we define $\varepsilon$-approximate truthfulness for a mechanism in both the Bayesian and frequentist paradigms.

*$\varepsilon$-Approximate Truthfulness:* All agents submitting truthfully ($f_i = I$), is an $\varepsilon$-approximate Nash equilibrium. In the Bayesian setting this means $\forall i \in [m], \forall f_i \in \mathcal{F}$

$$\mathbb{E}_{\mathcal{P} \sim \Pi}\left[\mathbb{E}_{\{X_i\}_{i=1}^m \sim \mathcal{P}}\left[L_i(\{I\}_{j=1}^m)\right]\right] \leq \mathbb{E}_{\mathcal{P} \sim \Pi}\left[\mathbb{E}_{\{X_i\}_{i=1}^m \sim \mathcal{P}}\left[L_i(f_i, \{I\}_{j \neq i})\right]\right] + \varepsilon.$$

In the frequentist setting this means $\forall i \in [m], \forall f_i \in \mathcal{F}$

$$\sup_{\mathcal{P} \in \mathcal{C}} \mathbb{E}_{\{X_i\}_{i=1}^m \sim \mathcal{P}}\left[L_i(\{I\}_{j=1}^m)\right] \leq \sup_{\mathcal{P} \in \mathcal{C}} \mathbb{E}_{\{X_i\}_{i=1}^m \sim \mathcal{P}}\left[L_i(f_i, \{I\}_{j \neq i})\right] + \varepsilon.$$

Algorithm 3 requires a similar non-degeneracy condition for MIB. In the Bayesian setting, the same condition given in Theorem 2 suffices. In the frequentist setting, we require that the class of distributions $\mathcal{C}$ is not solely comprised of distributions for which all of the feature map induced distributions are degenerate. The following theorem summarizes the main properties of Algorithm 3. We see that as the total amount of data increases, the approximate truthfulness parameter vanishes provided that the datasets $(X_i, W_i)$ and $Z_i$ are balanced.

**Theorem 3.** *The mechanism in Algorithm 3 is $\frac{1}{4}\left(\frac{1}{|X_i| + |W_i|} + \frac{1}{|Z_i|}\right)$-approximately truthful in both the Bayesian and frequentist settings. Moreover, if there is a feature $k \in [K]$, for which $\Pi$ is not degenerate, then Algorithm 3 satisfies MIB in the Bayesian setting. If it is not the case that $\mathcal{C} \subseteq \left\{\mathcal{P} \in \mathcal{M}_1(\mathcal{X}) : \forall k \in [K], \mathcal{P} \circ (\varphi^k)^{-1} \in \delta_x, x \in \mathbb{R}\right\}$ then Algorithm 3 satisfies MIB in the frequentist setting.*

Proposition 10 (Appendix D) gives, for both the Bayesian and frequentist settings, bounds on how an agent's expected loss decreases with the amount of data they submit, assuming all agents are truthful. Moreover, when the pushforward $\mathcal{P}^k = \mathcal{P} \circ \left( \varphi^k \right)^{-1}$ is a.s. continuous $\forall k \in [K]$, this proposition provides an exact expression for the expected loss in both the Bayesian and frequentist settings.

## 4 Applications to Data Sharing Problems

**1. A data marketplace for purchasing existing data.** Our first problem, studied by Chen et al. [10] is incentivizing agents to truthfully submit data using payments from a fixed budget $B$ in a Bayesian setting. Their mechanism requires the data distribution to have finite support or belong to the exponential family to ensure budget feasibility (payments do not exceed $B$) and individual rationality (agents receive non-negative payments). Our method removes these distributional assumptions.

In this setting, $m$ agents each posses a dataset $X_i = \{X_{i,1}, \ldots, X_{i,n_i}\}$ with points drawn i.i.d. from an unknown distribution $\mathcal{P}$ in a Bayesian model. A data analyst with budget $B$ wishes to purchase this data. Agents submit datasets $\{Y_i\}_{i=1}^m$ in return for payments $\{\pi_i(\{Y_i\}_i)\}_{i=1}^m$. Chen et al. [10], building on Kong and Schoenebeck [38], design a truthful mechanism based on log pairwise mutual information, but their payments can be unbounded, violating budget feasibility and individual rationality. We address this using Algorithm 2 to construct bounded payments satisfying truthfulness, individual rationality, and budget feasibility without distributional assumptions. Algorithm 4 (see Appendix A.1) implements this, and Proposition 2 guarantees these properties.

**2. A data marketplace to incentivize data collection at a cost.** The second problem, studied by Chen et al. [23], involves designing a data marketplace in which a buyer wishes to pay agents to collect data on her behalf *at a cost*. They study a Gaussian mean estimation problem in a frequentist setting. We study a simplified Bayesian version without assuming Gaussianity.

In a data marketplace mechanism, the interaction between the buyer and agents takes place as follows. First, each agent chooses how much data to collect, $n_i \in \mathbb{N}$, paying a *known* per-sample cost $c$, and obtains the dataset $X_i = \{X_{i,1}, \ldots, X_{i,n_i}\}$ with data drawn i.i.d. from an unknown $\mathcal{P} \sim \Pi$. They submit $Y_i = f_i(X_i)$ to the mechanism, and in return, receive a payment $\pi_i(\{Y_i\}_{i=1}^m)$ charged to the buyer. The buyer derives value $v : \mathbb{Z}_{\geq 0} \to \mathbb{R}_{\geq 0}$ from the total amount of truthful data received. An agent's utility is their expected payment minus collection cost $u_i^a = \mathbb{E}[\pi_i(\{Y_i\}_{i=1}^m)] - cn_i$, and the buyer's utility, when agents are truthful, is the valuation of the data received minus the expected sum of payments, $u^b = v\left(\sum_{i=1}^m |Y_i|\right) - \mathbb{E}\left[\sum_{i=1}^m \pi_i(\{Y_i\}_{i=1}^m)\right]$.

The goal of a data market mechanism is to incentivize agents to collect and truthfully report data. If not carefully designed, the mechanism may incentivize agents to fabricate data to earn payments without incurring collection costs, undermining market integrity and deterring buyers. To address this, we propose Algorithm 5 (see Appendix A.2), using Algorithm 2, which—unlike Chen et al. [23]—does not assume Gaussianity. Proposition 3 shows that, under a market feasibility condition, the mechanism is incentive compatible for agents and individually rational for buyers.

**3. Federated learning.** The third problem is a simple federated learning setting, similar to Karimireddy et al. [8], where agents share data to improve personalized models. Unlike their work, which assumes agents truthfully report collected data, we allow strategic misreporting.

Each of $m$ agents, possess a dataset $X_i = \{X_{i,1}, \ldots, X_{i,n_i}\}$ of points drawn i.i.d. in a Bayesian model, and have a valuation function $v_i : \mathbb{N} \to \mathbb{R}$ (increasing), quantifying the value of using a given amount of data for their machine learning task. Acting alone, an agent's utility is simply $v_i(|X_i|)$. When participating, the federated learning mechanism delopys a subset of the others' data submitted, $Z_i$, for agent $i$'s task based on the quality of their submission $f_i(X_i)$. This result in a valuation of $v_i(|Z_i|)$ when the others are truthful. Thus, an agent's utility when participating is defined as $u_i = \mathbb{E}[v_i(|Z_i|)]$. We propose Algorithm 6 (see Appendix A.3), based on Algorithm 2, which does not assume truthful reporting. Proposition 4 shows it is truthful and individually rational.

## 5 Experiments

**Synthetic experiments.** We consider two Bayesian models with conjugate priors (beta-Bernoulli and normal-normal) where the calculation of the conditional expectation in line 7 of Algorithm 1 is analytically tractable. In both setups, $\mathcal{X} = \mathbb{R}$ and we will use the method in Algorithm 1.

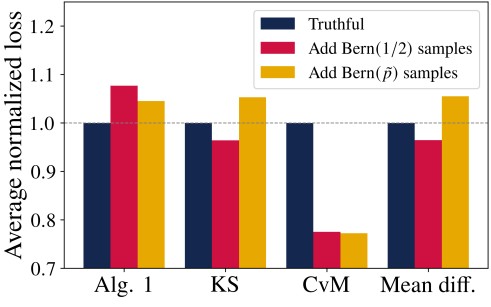

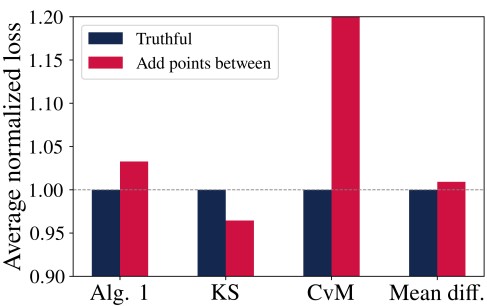

(a) Losses in the beta-bernoulli model

(b) Losses in the normal-normal model

Figure 2: (a): Losses when submitting truthfully, adding Bern $(1/2)$ samples, and adding Bern $(\tilde{p})$ samples in the beta-Bernoulli experiment. (b): Losses when submitting truthfully and adding fabricated data between adjacent pairs of true data points in the normal-normal experiment. In (b), the CvM bar for fabrication behavior extends to $\approx 1.6$. Losses for truthful submission in each method and subfigure are normalized to 1 (gray lines); values $< 1$ indicate fabrication improves performance, $> 1$ means it worsens. A truthful mechanism should yield losses above 1 for *all* fabrication behavior.

*Baselines:* We compare our mechanism to three standard two-sample tests, used here as losses: (1) the KS-test $\text{KS}(Y_i, Y_{-i}) = \sup_{t \in \mathbb{R}} |F_{Y_i}(t) - F_{Y_{-i}}(t)|$. (2) The CvM test (the direct version, not our adaptation): $\text{CvM}(Y_i, Y_{-i})$ (see (1)). (3) The mean difference (similar to the $t$-test): $\text{Mean-diff}(Y_i, Y_{-i}) = \left| \frac{1}{|Y_i|} \sum_{y \in Y_i} y - \frac{1}{|Y_{-i}|} \sum_{y \in Y_{-i}} y \right|$, which has been used to incentivize truthful reporting for normal mean estimation in a frequentist settings [7, 23].

*1) Beta-Bernoulli.* Our first model is a beta-Bernoulli Bayesian model with $p \sim \text{Beta}(2, 2)$ and then $X_{i,j}|p \sim \text{Bern}(p)$ i.i.d. We evaluate whether an agent can reduce their loss (increase rewards) by adding fabricated data to their submission. We consider two types of fabrication: (1) adding Bern $(1/2)$ samples and (2) estimating $p$ via $\tilde{p} = \frac{1}{|X_{i,j}|} \sum_{x \in X_{i,j}} x$ then adding Bern $(\tilde{p})$ samples. We compare this to an agent's loss when submitting truthfully, assuming in both cases that other agents are truthful. Fig. 2a shows average losses under Algorithm 1 and the three two-sample tests under truthful and non-truthful reporting. Under Algorithm 1, fabricated data always leads to higher loss, while the baselines yields lower loss under at least one fabrication strategy. Thus, the two-sample tests are susceptible to data fabrication whereas Algorithm 1 is not. Notably, Mean-diff, which is used in [7, 11], fails, showing their methods do not work beyond normal mean estimation settings.

*2) Normal-normal.* Our second experiment is a normal-normal Bayesian model, where $\mu \sim \mathcal{N}(0, 1)$ and then $X_{i,j}|\mu \sim \mathcal{N}(\mu, 1)$ i.i.d. Here, we fabricate data by inserting fake points in between real observations. Fig. 2b presents the results. Truthful reporting yields lower loss under Algorithm 1, CvM, and Mean-diff, while KS gives lower loss for fabrication, revealing its susceptibility.

**Language data.** Next, we evaluate our method and the above baselines on language data. For this, we use data from the SQuAD dataset [41], where each data point is a question about an article. We model the environment with $m = 20$ and $m = 100$ agents, where all agents have 2500 and 500 original data points respectively. We fabricate data by prompting Llama 3.2-1B-Instruct [26] to generate fake sentences based on the legitimate sentences that agent 1 has. We fabricate the same number of sentences in the original dataset. Agent 1 then submits the combined dataset, both true and fabricated, to the mechanism. We instantiate Algorithm 3 with feature maps obtained from the feature layer of the DistilBERT [42] encoder model, which corresponds to 768 features. We apply the baselines to the same set of features and take the average. We have provided additional details on the experimental set up and some true and fabricated sentences generated in Appendix B.1.

The results are presented in Table 1, showing that all methods perform well, obtaining a smaller loss for truthful submission when compared to fabricating. It is worth emphasizing that only our method is provably approximately truthful, and other methods may be susceptible to more sophisticated types of fabrication.

**Image data.** We perform a similar experiment on image data using the Oxford Flowers-102 dataset [43] dataset. where each data point is an image of a flower. We model the enviornment with $m = 5$ and $m = 47$, where all agents have roughly 1000 and 100 original data points respectively.

Table 1: An agent's average loss ($\pm$ the standard error) when reporting sentences truthfully/untruthfully, assuming the others are reporting truthfully. The experiments were run once assuming all agents had 500 sentences, then again assuming all agents had 2500 sentences. In each row the smaller loss is bolded.

| Sentences | Method | Avg. truthful loss | Avg. untruthful loss |
|---|---|---|---|
| 500 | Algorithm 3 | $\mathbf{0.0003} \pm 1.8 \cdot 10^{-5}$ | $0.0011 \pm 5.8 \cdot 10^{-5}$ |
| | KS-test | $\mathbf{0.0379} \pm 7.6 \cdot 10^{-4}$ | $0.0524 \pm 9.7 \cdot 10^{-4}$ |
| | CvM-test | $\mathbf{0.1547} \pm 9.2 \cdot 10^{-3}$ | $0.8598 \pm 4.8 \cdot 10^{-2}$ |
| | Mean diff. | $\mathbf{0.0043} \pm 2.5 \cdot 10^{-4}$ | $0.0095 \pm 3.4 \cdot 10^{-4}$ |
| 2500 | Algorithm 3 | $\mathbf{0.00003} \pm 3.3 \cdot 10^{-6}$ | $0.0005 \pm 7.1 \cdot 10^{-6}$ |
| | KS-test | $\mathbf{0.0127} \pm 2.4 \cdot 10^{-4}$ | $0.0309 \pm 1.2 \cdot 10^{-4}$ |
| | CvM-test | $\mathbf{0.1609} \pm 7.1 \cdot 10^{-3}$ | $3.2760 \pm 3.4 \cdot 10^{-2}$ |
| | Mean diff. | $\mathbf{0.0015} \pm 8.4 \cdot 10^{-5}$ | $0.0069 \pm 5.9 \cdot 10^{-5}$ |

We fabricate data by using Segmind Stable Diffusion-1B [25], a lightweight diffusion model, to generate fake images of flowers based on the legitimate pictures. We fabricate the same number of images that an agent possesses. Algorithm 3 is instantiated with 384 feature maps corresponding to the 384 nodes in the embedding layer of DeIT-small-distilled [44], a small vision transformer. As above, we apply the baselines to the same set of features and take the average. Additional details on the experimental set up can be found in Appendix B.2.

Table 2 shows that, similar to text, all methods perform well, truthful submission leads to a lower loss compared to the fabrication procedure detailed above.

Table 2: An agent's average loss ($\pm$ the standard error) when reporting images truthfully/untruthfully, assuming the others are reporting truthfully. The experiments were run once assuming agent 1 had 100 images, then again assuming agent 1 had 1000 images. The 4,612 images in the test set of [43] were used to represent the data submitted by other agents. In each row the smaller loss is bolded.

| Images | Method | Avg. truthful loss | Avg. untruthful loss |
|---|---|---|---|
| 100 | Algorithm 3 | $\mathbf{0.0015} \pm 3.2 \cdot 10^{-5}$ | $0.0040 \pm 1.2 \cdot 10^{-4}$ |
| | KS-test | $\mathbf{0.0833} \pm 4.2 \cdot 10^{-4}$ | $0.0993 \pm 1.3 \cdot 10^{-3}$ |
| | CvM-test | $\mathbf{0.1491} \pm 2.6 \cdot 10^{-3}$ | $0.7730 \pm 2.0 \cdot 10^{-2}$ |
| | Mean diff. | $\mathbf{0.0462} \pm 1.0 \cdot 10^{-3}$ | $0.0953 \pm 1.1 \cdot 10^{-3}$ |
| 1000 | Algorithm 3 | $\mathbf{0.0002} \pm 3.7 \cdot 10^{-6}$ | $0.0032 \pm 2.9 \cdot 10^{-5}$ |
| | KS-test | $\mathbf{0.0290} \pm 2.1 \cdot 10^{-4}$ | $0.0738 \pm 2.7 \cdot 10^{-4}$ |
| | CvM-test | $\mathbf{0.1458} \pm 3.5 \cdot 10^{-3}$ | $4.5478 \pm 3.0 \cdot 10^{-2}$ |
| | Mean diff. | $\mathbf{0.0157} \pm 5.2 \cdot 10^{-4}$ | $0.0896 \pm 3.2 \cdot 10^{-4}$ |

## 6 Conclusion

We study designing mechanisms that incentivize truthful data submission while rewarding agents for contributing more data. In the Bayesian setting, we propose a mechanism that satisfies these goals under a mild non-degeneracy condition on the prior. We additionally develop a prior-agnostic variant that applies in both Bayesian and frequentist settings. We illustrate the practical utility of our mechanisms by revisiting data sharing problems studied in prior work, relaxing their technical assumptions, and validating our approach through experiments on synthetic and real-world datasets.

*Limitations.* The mechanisms in §2 rely on Bayesian posterior computations, which may be computationally expensive for complex priors. We also require specifying feature maps that effectively represent the data. While this offers flexibility for the mechanism designer to select application-specific features, there is no universally optimal way to choose them.

## Acknowledgments and Disclosure of Funding

This work was partially supported by NSF grant IIS-2441796.

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

# A Omitted application algorithms

## A.1 A data marketplace for purchasing existing data

Recall the problem setup from §4. Below we provide a short algorithm that incentivizes agents to truthfully report their data, $\{X_i\}_{i=1}^m$, using payments. The idea is to use Algorithm 2 to quantify the quality of an agent's submission and, based on it, determine what fraction of the budget to pay them.

**Definition 3.** *We say an algorithm is budget feasible if the sum of the payments never exceeds the budget ($\sum_{i=1}^m \pi_i \leq B$), and individually rational (for participants) if the payments are always nonnegative ($\forall i \in [m], \pi_i \geq 0$).*

---
**Algorithm 4** A data marketplace for purchasing existing data
---
1: **Input parameters**: A prior $\Pi$ over the set of $\mathcal{X}$-valued distributions, feature maps $\{\varphi^k\}_{k=1}^K$.
2: Receive datasets $Y_1, \ldots, Y_m$ from the agents.
3: Execute Algorithm 2 with $\{Y_i\}_{i=1}^m$, $\Pi$, $\{\varphi^k\}_{k=1}^K$, to obtain the loss $L_i \in [0,1]$ for agent $i$.
4: Pay agent $i$: $\pi_i (\{Y_i\}_{i=1}^m) = \frac{B}{m} (1 - L_i (\{Y_i\}_{i=1}^m))$.
---

**Proposition 2.** *Algorithm 4 is truthful, individually rational, and budget feasibility.*

*Proof.* Since $L_i \in [0,1]$, we have $0 \leq \pi_i \leq \frac{B}{m}$, so it immediately follows that Algorithm 4 is both individually rational for the agents and budget feasible. For truthfulness, notice that for any $f_i \in \mathcal{F}$, we can appeal to Theorem 2 to get

$$\mathbb{E}\left[\pi_i\left(f_i, \{I\}_{j\neq i}\right)\right] = \frac{B}{m}\left(1 - \mathbb{E}\left[L_i\left(f_i, \{I\}_{j\neq i}\right)\right]\right)$$
$$\leq \frac{B}{m}\left(1 - \mathbb{E}\left[L_i\left(\{I\}_{i=1}^m\right)\right]\right)$$
$$= \mathbb{E}\left[\pi_i\left(\{I\}_{i=1}^m\right)\right].$$

Therefore, Algorithm 4 is also truthful, as agents maximize their expected payment when submitting truthfully. $\square$

## A.2 A data marketplace to incentivize data collection at a cost

Recall the problem setup from §4, which is a simplified version of the problem studied by [23]. Our setting does not subsume [23], as they allow for agents to have varying collection costs, study a frequentist setting (whereas we consider a Bayesian setting), and derive payments that are easy to compute. We now motivate a solution to our simplified setting.

To facilitate data sharing between a buyer and agents, a mechanism must first determine how much data agents should be asked to collect based on the cost of data collection $c$, and the buyer's valuation function $v$. To do this, suppose that the buyer could collect data himself. In this case, he would choose to collect $n^{\text{OPT}} := \text{argmax}_{n \in \mathbb{N}} (v(n) - cn)$ points to maximize his utility. However, as he cannot, when there are $m$ agents, the mechanism will ask each of them to collect $\frac{n^{\text{OPT}}}{m}$ points on his behalf in exchange for payments.

An important detail is that for the marketplace to be feasible, an agent's expected payment must outweigh the cost of data collection. This requirement is reflected in the first technical condition in Proposition 3, which at a high level says that the change in an agents expected payment with respect to $n_i$, when collecting $\frac{n^{\text{OPT}}}{m}$ points, is at least $c$. This can be thought of requiring that the derivative with repect to $n_i$, of the expected payment at $\frac{n^{\text{OPT}}}{m}$, be at least $c$. We also assume that $\Pi$ is not degnerate for all the features and there are deminishing returns for collecting and submitting more data under Algorithm 2.

When these condition holds, Proposition 3 shows that it is individually rational for a buyer to participate in the marketplace, and in agents' best interest to collect $\frac{n^{\text{OPT}}}{m}$ points and submit them truthfully.

The idea of Algorithm 5 is to determine what fraction of $\frac{v(n^{\mathrm{OPT}})}{m}$ to pay agent $i$ based on the quality of her submission, as measured by $L_i$.

---

**Algorithm 5** A data marketplace to incentivize data collection at a cost

---

1: **Input parameters**: A prior $\Pi$ over the set of $\mathcal{X}$-valued distributions, feature maps $\{\varphi^k\}_{k=1}^K$.
2: Receive datasets $Y_1, \ldots, Y_m$ from the agents.
3: Execute Algorithm 2 with $\{Y_i\}_{i=1}^m$, $\Pi$, $\{\varphi^k\}_{k=1}^K$, to obtain the loss $L_i \in [0,1]$ for agent $i$.
4: Pay agent $i$: $\pi_i\left(\{Y_i\}_{i=1}^m\right) = \frac{v(n^{\mathrm{OPT}})}{m}\left(1 - \alpha L_i\right)$ where $\alpha$ is given in Definition 4.
5: Charge the buyer: $p\left(\{Y_i\}_{i=1}^m\right) = \sum_{i=1}^m \pi_i\left(\{Y_i\}_{i=1}^m\right)$.

---

**Definition 4.** *For Algorithm 5 we introduce notation for the change in an agent's expected payment when collecting and submitting one more data point truthfully, assuming others are truthful:*

$$\frac{\partial}{\partial n_i}\mathbb{E}\left[L_i\left(n_i, n_{-i}\right)\right] := \mathbb{E}\left[L_i\left(\left(n_i+1, n_{-i}\right), \{I\}_{i=1}^m\right)\right] - \mathbb{E}\left[L_i\left(\left(n_i, n_{-i}\right), \{I\}_{i=1}^m\right)\right].$$

*When $\Pi$ is not degenerate $\forall k \in [K]$, $\frac{\partial}{\partial n_i}\mathbb{E}\left[L_i\left(\left\{\frac{n^{\mathrm{OPT}}}{m}\right\}_{i=1}^m\right)\right] < 0$ (by Theorem 2) and we define*

$$\alpha := -\frac{cm}{\frac{\partial}{\partial n_i}\mathbb{E}\left[L_i\left(\left\{\frac{n^{\mathrm{OPT}}}{m}\right\}_{i=1}^m\right)\right]v(n^{\mathrm{OPT}})}.$$

**Proposition 3.** *Suppose that the following technical conditions are satisfied in Algorithm 5:*

$$\frac{v\left(n^{\mathrm{OPT}}\right)}{m}\left(-\frac{\partial}{\partial n_i}\mathbb{E}\left[L_i\left(\left\{\frac{n^{\mathrm{OPT}}}{m}\right\}_{i=1}^m\right)\right]\right) \geq c,$$

*$\Pi$ is not degenerate $\forall k \in [K]$, and $-\frac{\partial}{\partial n_i}\mathbb{E}\left[L_i\left(n_i, n_{-i}\right)\right]$ is decreasing in $n_i$.*

*Then, the strategy profile $\left\{\left(\frac{n^{\mathrm{OPT}}}{m}, I\right)\right\}_{i=1}^m$ is individually rational for the buyer, i.e.*

$$u^b\left(\left\{\left(\frac{n^{\mathrm{OPT}}}{m}, I\right)\right\}_{i=1}^m\right) \geq 0$$

*and incentive compatible for the agents, i.e. for any $n_i \in \mathbb{N}$, $f_i \in \mathcal{F}$,*

$$u_i^a\left(\left(n_i, f_i\right), \left\{\left(\frac{n^{\mathrm{OPT}}}{m}, I\right)\right\}_{j \neq i}\right) \leq u_i^a\left(\left\{\left(\frac{n^{\mathrm{OPT}}}{m}, I\right)\right\}_{i=1}^m\right).$$

*Proof.* We start with individual rationality for the buyer. Notice that if the inequality holds then we have

$$\alpha = \frac{cm}{-\frac{\partial}{\partial n_i}\mathbb{E}\left[L_i\left(\left\{\frac{n^{\mathrm{OPT}}}{m}\right\}_{i=1}^m\right)\right]v(n^{\mathrm{OPT}})} \leq \frac{cm}{\frac{cm}{v(n^{\mathrm{OPT}})}v(n^{\mathrm{OPT}})} = 1$$

so $\alpha \in (0,1]$. Since $L_i \in [0,1]$, this implies that

$$\pi_i\left(\{Y_i\}_{i=1}^m\right) = \frac{v(n^{\mathrm{OPT}})}{m}\left(1 - \alpha L_i\right) \leq \frac{v(n^{\mathrm{OPT}})}{m}$$

so summing over the payments to all agents we find

$$p\left(\{Y_i\}_{i=1}^m\right) = \sum_{i=1}^m \pi_i\left(\{Y_i\}_{i=1}^m\right) \leq v(n^{\mathrm{OPT}}).$$

Therefore, the strategy profile $\left\{\left(\frac{n^{\mathrm{OPT}}}{m}, I\right)\right\}_{i=1}^m$ is individually rational for the buyer since

$$u^b\left(\left\{\left(\frac{n^{\mathrm{OPT}}}{m}, I\right)\right\}_{i=1}^m\right) = v(n^{\mathrm{OPT}}) - \mathbb{E}\left[p\left(\{Y_i\}_{i=1}^m\right)\right] \geq 0.$$

We now prove incentive compatibility for the agents in two parts. First we show that regardless of how much data an agent has collected, it is best for her to submit it truthfully when others follow the recommended strategy profile $\left\{\left(\frac{n^{\mathrm{OPT}}}{m}, I\right)\right\}_{j\neq i}$. Second, we show that $\frac{v(n^{\mathrm{OPT}})}{m}$ is the optimal amount of data to collect based on our choice of $\alpha$.

Fix $n_i$. Unpacking the definition of an agent's utility and applying Theorem 2 we have

$$
u_i^a\left((n_i, f_i), \left\{\left(\frac{n^{\mathrm{OPT}}}{m}, I\right)\right\}_{j\neq i}\right)
$$

$$
= \mathbb{E}\left[\pi_i\left((n_i, f_i), \left\{\left(\frac{n^{\mathrm{OPT}}}{m}, I\right)\right\}_{j\neq i}\right)\right] - cn_i
$$

$$
= \frac{v\left(n^{\mathrm{OPT}}\right)}{m}\left(1 - \alpha\mathbb{E}\left[L_i\left((n_i, f_i), \left\{\left(\frac{n^{\mathrm{OPT}}}{m}, I\right)\right\}_{j\neq i}\right)\right]\right) - cn_i
$$

$$
\leq \frac{v\left(n^{\mathrm{OPT}}\right)}{m}\left(1 - \alpha\mathbb{E}\left[L_i\left((n_i, I), \left\{\left(\frac{n^{\mathrm{OPT}}}{m}, I\right)\right\}_{j\neq i}\right)\right]\right) - cn_i
$$

$$
= \mathbb{E}\left[\pi_i\left((n_i, I), \left\{\left(\frac{n^{\mathrm{OPT}}}{m}, I\right)\right\}_{j\neq i}\right)\right] - cn_i
$$

$$
= u_i^a\left((n_i, I), \left\{\left(\frac{n^{\mathrm{OPT}}}{m}, I\right)\right\}_{j\neq i}\right).
$$

This means that regardless of how much data agent $i$ collects, it is best for them to submit it truthfully. For the second part we now assume $\{f_i\}_{i=1}^m = \{I\}_{i=1}^m$ and $n_{-i} = \left\{\frac{n^{\mathrm{OPT}}}{m}\right\}_{i=1}^m$ so for convenience we omit writing the dependence on these parts of the strategy profile for random variables.

Notice that since $u_i^a(n_i)$ is concave, the optimal amount of data for agent $i$ to collect and submit is the smallest $n_i \in \mathbb{N}$ such that

$$
u_i^a(n_i + 1) \leq u_i^a(n_i)
$$

i.e. the point at which the marginal increase in payment no longer offsets the collection cost of an additional point. By the definition of agent utilities and our choice of $\alpha$ we see

$$
u_i^a(n_i + 1) \leq u_i^a(n_i) \iff \left(\frac{v(n^{\mathrm{OPT}})}{m}(1 - \alpha\mathbb{E}[L_i(n_i + 1)]) - c(n_i + 1)\right)
$$

$$
\leq \left(\frac{v(n^{\mathrm{OPT}})}{m}(1 - \alpha\mathbb{E}[L_i(n_i)]) - c(n_i)\right)
$$

$$
\iff -\frac{\partial}{\partial n_i}\mathbb{E}[L_i(n_i)] \leq \frac{cm}{\alpha v(n^{\mathrm{OPT}})}
$$

$$
\iff -\frac{\partial}{\partial n_i}\mathbb{E}[L_i(n_i)] \leq -\frac{\partial}{\partial n_i}\mathbb{E}\left[L_i\left(\frac{n^{\mathrm{OPT}}}{m}\right)\right].
$$

This implies that $\frac{n^{\mathrm{OPT}}}{m}$ is the optimal amount of data to collect since $-\frac{\partial}{\partial n_i}\mathbb{E}[L_i(n_i, n_{-i})]$ is decreasing in $n_i$. Putting both parts together we find that for any $n_i \in \mathbb{N}$, $f_i \in \mathcal{F}$,

$$
u_i^a\left((n_i, f_i), \left\{\left(\frac{n^{\mathrm{OPT}}}{m}, I\right)\right\}_{j\neq i}\right) \leq u_i^a\left((n_i, I), \left\{\left(\frac{n^{\mathrm{OPT}}}{m}, I\right)\right\}_{j\neq i}\right)
$$

$$
\leq u_i^a\left(\left\{\left(\frac{n^{\mathrm{OPT}}}{m}, I\right)\right\}_{i=1}^m\right)
$$

so we have incentive compatibility for the agents.

$\square$

## A.3 Federated learning

Recall the problem setup from §4. For convenience we assume that $\forall i \in [m], |X_i| < \sum_{j \neq i} |X_j|$.

The idea of Algorithm 6 is to determine how much of the others' data agent $i$ should receive for her task based on the quality of her submission, as measured by $L_i$.

---

**Algorithm 6** Federated learning

---

1: **Input parameters**: A prior $\Pi$ over the set of $\mathcal{X}$-valued distributions, feature maps $\{\varphi^k\}_{k=1}^K$.
2: Receive datasets $Y_1, \ldots, Y_m$ from the agents.
3: Execute Algorithm 2 with $\{Y_i\}_{i=1}^m$ $\Pi$, $\{\varphi^k\}_{k=1}^K$, to obtain the loss $L_i \in [0,1]$ for agent $i$.
4: **for** each agent $i \in [m]$:
5: $\quad T_i \leftarrow v_i(|Y_{-i}|), \quad \alpha \leftarrow \left(\frac{1}{2} - \frac{v_i(|X_i|)}{2T_i}\right) \frac{1}{\mathbb{E}[L_i(\{I\}_{i=1}^m)]}$
6: $\quad z_i \leftarrow v_i^{-1}\left((1 - \alpha L_i)\, T_i\right)$
7: $\quad$ Deploy $Z_i$, a random subset of $Y_{-i}$ of size $z_i$ for agent $i$'s machine learning task.

---

In Algorithm 6 and Proposition 4 we assume that $\mathbb{E}[L_i(\{I\}_{i=1}^m)] > 0$ which ensures $\alpha$ is well defined by rulling out trivial data sharing problems.

**Proposition 4.** *Suppose that* $\forall i \in [m], |X_i| < \sum_{j \neq i} |X_j|$. *Then Algorithm 6 is truthful and individually rational.*

*Proof.* Fix $f_i \in \mathcal{F}$. Unpacking the definition of an agent's utility and applying Theorem 2, we have

$$
\begin{aligned}
u_i\left(f_i, \{I\}_{j \neq i}\right) &= \mathbb{E}\left[v_i\left(z_i\left(f_i, \{I\}_{j \neq i}\right)\right)\right] \\
&= \mathbb{E}\left[v_i\left(v_i^{-1}\left(\left(1 - \alpha L_i\left(f_i, \{I\}_{j \neq i}\right)\right) T_i\right)\right)\right] \\
&= T_i - T_i \alpha \mathbb{E}\left[L_i\left(f_i, \{I\}_{j \neq i}\right)\right] \\
&\leq T_i - T_i \alpha \mathbb{E}\left[L_i\left(\{I\}_{i=1}^m\right)\right] \\
&= u_i\left(\{I\}_{i=1}^m\right).
\end{aligned}
$$

Therefore, Algorithm 6 is truthful. For individual rationality, notice that by the definition of $\alpha$ and the assumption that $|X_i| < |X_{-i}|$ $\left(\text{and thus } v_i(X_i) < v_i(X_{-i})\right)$, we have

$$
\begin{aligned}
u_i\left(\{I\}_{i=1}^m\right) &= T_i - T_i \alpha \mathbb{E}\left[L_i\left(\{I\}_{i=1}^m\right)\right] \\
&= T_i - T_i\left(\frac{1}{2} - \frac{v_i(|X_i|)}{2T_i}\right) \\
&= \frac{v_i(|X_{-i}|)}{2} + \frac{v_i(|X_i|)}{2} \\
&> v_i(|X_i|).
\end{aligned}
$$

Therefore, agent $i$ is better off participating in Algorithm 6 than working alone so individual rationality is satisfied. $\qquad\square$

## B Extended experimental results and details

### B.1 Text based experiments

Our first real world experiment supposes that agents possess and wish to share text data drawn from a common distribution. To simulate this text distribution, we use data from the SQuAD[1] dataset [41] which contains 100,000 questions generated by providing crowdworkers with snippets from Wikipedia articles and asking them to formulate questions based on the snippet's content. We simulate

---

data sharing when $m = 20$ and $m = 100$, where agents have 2,500 and 500 original data points respectively.

[1]This work uses the Stanford Question Answering Dataset (SQuAD), which is licensed under the Creative Commons Attribution-ShareAlike 4.0 International (CC BY-SA 4.0) license.

When agents are truthful, they simply submit their sentences to the mechanism (Algorithm 3). However, an untruthful agent can fabricate fake sentences to augment their dataset with in hopes of achieving a lower loss. We consider when agents attempt to do this using an LLM (Llama 3.2-1B-Instruct [26]) by prompting it to produce authentic looking sentences based on legitimate sentences Fig. 3 shows an example of the prompting and Table 4 shows examples of the LLM-generated sentences. For consistency, we filter out duplicates and any outputs not ending in a question mark.

---

**Prompt**

Generate five new questions that follow the same style as the examples below.
Each question should be separated by a newline.

According to Southern Living, what are the three best restaurants in Richmond?
When did the Arab oil producers lift the embargo?
Complexity classes are generally classified into what?
About how many acres is Pippy Park?
Which BYU station offers content in both Spanish and Portuguese?

---

Figure 3: Pictured above is an example prompt fed into Llama 3.2-1B-Instruct as part of an untruthful agent's submission function to generate fabricated text data. The agent uses their five questions drawn from the SQuAD to fabricate similar five additonal questions.

Table 3: Comparison of SQuAD questions versus LLM-generated fabrications.

| SQuAD questions (Real) | LLM-generated questions (Fabricated) |
| --- | --- |
| Which tribe did Temüjin move in with at nine years of age? | What percentage of the population of France lived in urban areas as of 2019? |
| What is the most widely known fictional work from the Islamic world? | The term *solar eclipse* refers to what phenomenon? |
| New Delhi played host to what major athletic competition in 2010? | Is it true that the first computer bug was an actual insect? |
| Why did the FCC reject systems such as MUSE? | How many Earth years is Neptune's south pole exposed to the Sun? |
| Along with the philosophies of music and art, what field of philosophy studies emotions? | Military spending based on conventional threats has been dismissed as what? |

To incentivize truthful submission, we instantiate Algorithm 3 with 768 feature maps corresponding to the 768 nodes in the embedding layer of DistilBERT [42], a lightweight encoder model distilled from the encoder transformer model Bert [45]. For simplicity, we chose the split map $\psi(n) = 0$. As a point of comparison, we also apply the KS, CvM, and Mean diff. tests (described in §5), now to the 768 node feature space.

Our results comparing the average loss agent $i$ receives when submitting truthfully/untruthfully, under the four methods, over five runs, are given in Table 1. We see that under all of the methods truthful submission results in a lower average loss than untruthful submission.

### B.2 Image based experiments

Our second experiment supposes that agents wish to share image data from a common distribution. To simulate this image distribution, we use data from the Oxford Flowers-102 dataset [43], which contains 6,149 images across 102 flower categories. We simulate data sharing when an agent 1 has 100 and 1,000 images as data points. We use the test dataset of [43], which consists of 4,612 images, to represent authentic data submitted by the other agents. In the two scenarios, this roughly corresponds to $m = 47$ agents each with 100 images and $m = 5$ agents each with 1000 images.

When agent are untruthful, they may fabricate images using a diffusion model to augment their dataset. We consider when agents use Segmind Stable Diffusion-1B [25], a lightweight diffusion

model, to do this. More specifically, for each sampled image, we use it in conjunction with the prompts and parameters in Table 4 to generate an additional fabricated image.

Table 4: Parameters and prompts used for Segmind Stable Diffusion-1B to generate the fabricated images. Here cls_name is replaced with the type of flower being generated.

| Parameter | Value |
| --- | --- |
| Text Prompt | Photorealistic photograph of a single {cls_name}, realistic colors, natural lighting, high detail, sharp focus on petals. Another unique photo of the same flower species. |
| Negative Prompt | oversaturated, highly saturated, neon colors, garish colors, vibrant colors, illustration, painting, drawing, sketch, cartoon, anime, unrealistic, blurry, low quality, text, watermark, signature, border, frame, multiple flowers |
| Strength | 0.7 |
| Guidance Scale | 6 |
| Num. Inference Steps | 50 |

To discourage fabrication, Algorithm 3 is now instantiated with 384 feature maps corresponding to the 384 nodes in the embedding layer of DeIT-small-distilled [44], a small vision transformer. For simplicity, we chose the split map $\psi(n) = 0$. As a point of comparison, we again apply the KS, CvM, and Mean diff. tests (described in §5), now to the 384 node feature space. Our results comparing the average loss agent $i$ receives when submitting truthfully/untruthfully, under the four methods, over five runs, can be found in Table 2.

We find that truthful reporting outperforms untruthful reporting for all methods, demonstrating they are not susceptible to diffusion based fabrication.

## C   Results and proofs omitted from Section 2

### C.1   Results and proofs omitted from Subsection 2.1

**Theorem 1.** *The mechanism in Algorithm 1 satisfies truthfulness. Moreover, when $\Pi$ is not degenerate, then Algorithm 1 also satisfies MIB.*

*Proof.* For truthfulness we refer to Proposition 5.

For $n = (n_1, \ldots, n_m) \in \mathbb{N}^m$, let $L_i(n, \{I\}_{i=1}^m)$ denote the value of $L_i$ in Algorithm 1 when agent $j \in [m]$ has $n_j$ data points and agents use $\{I\}_{i=1}^m \in \mathcal{F}^m$. Proposition 6 tells us that

$$\mathbb{E}\left[L_i\left(n, \{I\}_{i=1}^m\right)\right] - \mathbb{E}\left[L_i\left(n + e_i, \{I\}_{i=1}^m\right)\right] = \mathbb{E}\left[\left(\mathbb{E}\left[F_{Z_i}(T_i) | \mathcal{G}_{n_i}\right] - \mathbb{E}\left[F_{Z_i}(T_i) | \mathcal{G}_{n_i+1}\right]\right)^2\right].$$

where $\mathcal{G}_j = \sigma\left(X_{i,1}, \ldots X_{i,j}, T_i\right)$. Also notice that

$$P\left(Z_{i,1} \leq T_i | X_{i,1}, \ldots, X_{i,n_i}, T_i\right) = \mathbb{E}\left[F_{Z_i}(T_i) | \mathcal{G}_{n_i}\right],$$
$$P\left(Z_{i,1} \leq T_i | X_{i,1}, \ldots, X_{i,n_i+1}, T_i\right) = \mathbb{E}\left[F_{Z_i}(T_i) | \mathcal{G}_{n_i+1}\right].$$

By definition $\Pi$ being non-degenerate means that the conditional probabilities are not almost surely equal. This implies that

$$\mathbb{E}\left[L_i\left(n, \{I\}_{i=1}^m\right)\right] - \mathbb{E}\left[L_i\left(n + e_i, \{I\}_{i=1}^m\right)\right] > 0$$

so the MIB property is satisfied.

$\square$

**Proposition 5.** *Let $L_i\left(\{f_i\}_{i=1}^m\right)$ denote the value of $L_i$ in Algorithm 1 when all agents use $\{f_i\}_{i=1}^m \in \mathcal{F}^m$. Then, for any $f_i \in \mathcal{F}$, $\mathbb{E}\left[L_i\left(\{I\}_{i=1}^m\right)\right] \leq \mathbb{E}\left[L_i\left(f_i, \{I\}_{j\neq i}\right)\right]$.*

*Proof.* By definition $\mathbb{E}\left[F_{Z_i}(T_i)\,\middle|\,X_{i,1},\ldots,X_{i,|Y_i|},T_i\right]$ is $\left(X_{i,1},\ldots,X_{i,|Y_i|},T_i\right)$-measurable, so there exists a measurable function $g:\mathbb{R}^{|Y_i|+1}\to\mathbb{R}$ such that

$$g\left(X_{i,1},\ldots,X_{i,|Y_i|},T_i\right)=\mathbb{E}\left[F_{Z_i}(T_i)\,\middle|\,X_{i,1},\ldots,X_{i,|Y_i|},T_i\right].$$

The conditional expectation $\mathbb{E}\left[F_{Z_i}(T_i)\,\middle|\,X_{i,1}=Y_{i,1},\ldots,X_{i,|Y_i|}=Y_{i,|Y_i|},T_i\right]$ is shorthand for $g\left(Y_{i,1},\ldots,Y_{i,|Y_i|},T_i\right)$. Since we assume $f_i$ is measurable, $\left(Y_{i,1},\ldots,Y_{i,|Y_i|}\right)=f_i\left(X_{i,1},\ldots,X_{i,n_i}\right)$ is $\left(X_{i,1},\ldots,X_{i,n_i}\right)$-measurable. Therefore, we know that

$$g\left(Y_{i,1},\ldots,Y_{i,|Y_i|},T_i\right)=\mathbb{E}\left[F_{Z_i}(T_i)\,\middle|\,X_{i,1}=Y_{i,1},\ldots,X_{i,|Y_i|}=Y_{i,|Y_i|},T_i\right]$$

is $\left(X_{i,1},\ldots,X_{i,n_i},T_i\right)$-measurable. This lets us apply Lemma 5 to get

$$\begin{aligned}
\mathbb{E}\left[L_i\left(f_i,\{I\}_{j\neq i}\right)\right] &= \mathbb{E}\left[\left(\mathbb{E}\left[F_{Z_i}(T_i)\,\middle|\,X_{i,1}=Y_{i,1},\ldots,X_{i,|Y_i|}=Y_{i,|Y_i|},T_i\right]-F_{Z_i}(T_i)\right)^2\right]\\
&\geq \mathbb{E}\left[\left(\mathbb{E}\left[F_{Z_i}(T_i)\,\middle|\,X_{i,1},\ldots,X_{i,n_i},T_i\right]-F_{Z_i}(T_i)\right)^2\right]\\
&= \mathbb{E}\left[L_i\left(\{I\}_{i=1}^m\right)\right].
\end{aligned}$$

$\square$

**Proposition 6.** *For $n=(n_1,\ldots,n_m)\in\mathbb{N}^m$ let $L_i\left(n,\{I\}_{i=1}^m\right)$ denote the value of $L_i$ in Algorithm 1 when agent $j\in[m]$ has $n_j$ data points and agents use $\{I\}_{i=1}^m\in\mathcal{F}^m$. Then*

$$\mathbb{E}\left[L_i\left(n,\{I\}_{i=1}^m\right)\right]-\mathbb{E}\left[L_i\left(n+e_i,\{I\}_{i=1}^m\right)\right]=\mathbb{E}\left[\left(\mathbb{E}\left[F_{Z_i}(T_i)\,\middle|\,\mathcal{G}_{n_i}\right]-\mathbb{E}\left[F_{Z_i}(T_i)\,\middle|\,\mathcal{G}_{n_i+1}\right]\right)^2\right]$$

*where $\mathcal{G}_j=\sigma\left(X_{i,1},\ldots X_{i,j},T_i\right)$.*

*Proof.* For convenience define $U=F_{Z_i}(T_i)$ and $V=\mathbb{E}\left[U|\mathcal{G}_{n_i}\right]$. By the definition of $L_i$ in Algorithm 1 and conditional variance we have

$$\begin{aligned}
\mathbb{E}\left[L_i\left(n,\{I\}_{i=1}^m\right)\right] &= \mathbb{E}\left[\left(\mathbb{E}\left[F_{Z_i}(T_i)\,\middle|\,X_{i,1},\ldots,X_{i,n_i},T_i\right]-F_{Z_i}(T_i)\right)^2\right]\\
&= \mathbb{E}\left[(U-V)^2\right]\\
&= \mathbb{E}\left[\mathbb{E}\left[(U-V)^2\,\middle|\,\mathcal{G}_{n_i}\right]\right]\\
&= \mathbb{E}\left[\mathrm{Var}\left(U|\mathcal{G}_{n_i}\right)\right].
\end{aligned}$$

Similarly we have

$$\mathbb{E}\left[L_i\left(n+e_i,\{I\}_{i=1}^m\right)\right]=\mathbb{E}\left[\mathrm{Var}\left(U|\mathcal{G}_{n_i+1}\right)\right].$$

Let $Y=\mathbb{E}\left[U|\mathcal{G}_{n_i+1}\right]$. We can now appeal to Lemma 4 to get

$$\begin{aligned}
\mathbb{E}\left[L_i\left(n,\{I\}_{i=1}^m\right)\right]-\mathbb{E}\left[L_i\left(n+e_i,\{I\}_{i=1}^m\right)\right] &= \mathbb{E}\left[\mathrm{Var}\left(U|\mathcal{G}_{n_i}\right)\right]-\mathbb{E}\left[\mathrm{Var}\left(U|\mathcal{G}_{n_i+1}\right)\right]\\
&= \mathbb{E}\left[\mathrm{Var}\left(Y|\mathcal{G}_{n_i}\right)\right].
\end{aligned}$$

Using the tower property gives

$$\begin{aligned}
\mathbb{E}\left[\mathrm{Var}\left(Y|\mathcal{G}_{n_i}\right)\right] &= \mathbb{E}\left[\mathbb{E}\left[\left(Y-\mathbb{E}\left[Y|\mathcal{G}_{n_i}\right]\right)^2\,\middle|\,\mathcal{G}_{n_i}\right]\right]\\
&= \mathbb{E}\left[\left(\mathbb{E}\left[U|\mathcal{G}_{n_i}\right]-\mathbb{E}\left[U|\mathcal{G}_{n_i+1}\right]\right)^2\right]\\
&= \mathbb{E}\left[\left(\mathbb{E}\left[F_{Z_i}(T_i)\,\middle|\,\mathcal{G}_{n_i}\right]-\mathbb{E}\left[F_{Z_i}(T_i)\,\middle|\,\mathcal{G}_{n_i+1}\right]\right)^2\right].
\end{aligned}$$

$\square$

**Proposition 1.** *Let $L_i\left(\{I\}_{i=1}^m\right)$ denote the value of $L_i$ when agents are truthful in Algorithm 1. Then, $0\leq\mathbb{E}\left[L_i\left(\{I\}_{i=1}^m\right)\right]\leq\frac{1}{4}\left(\frac{1}{|X_i|}+\frac{1}{|Z_i|}\right)$. Moreover, when $\Pi$ is a prior over the set of continuous $\mathbb{R}$-valued distributions, $\frac{1}{6|Z_i|}\leq\mathbb{E}\left[L_i\left(\{I\}_{i=1}^m\right)\right]\leq\frac{1}{6}\left(\frac{1}{|X_i|}+\frac{1}{|Z_i|}\right)$.*

*Proof.* Since $F_{X_i}(T_i)$ is $\sigma(X_{i,1}, \ldots, X_{i,n_i}, T_i)$-measurable, Lemma 5 tells us that

$$\mathbb{E}\left[L_i\left(\{I\}_{i=1}^m\right)\right] = \mathbb{E}\left[\left(\mathbb{E}\left[F_{Z_i}(T_i)\,|\,X_{i,1}, \ldots, X_{i,n_i}, T_i\right] - F_{Z_i}(T_i)\right)^2\right]$$

$$\leq \mathbb{E}\left[\left(F_{X_i}(T_i) - F_{Z_i}(T_i)\right)^2\right].$$

Now we can condition on $\mathcal{P}$ apply the first part of Lemma 2 to the inner expectation to get

$$\mathbb{E}\left[\left(F_{X_i}(T_i) - F_{Z_i}(T_i)\right)^2\right] = \mathbb{E}_{\mathcal{P}}\left[\mathbb{E}_{X_i, Z_i, T_i}\left[\left(F_{X_i}(T_i) - F_{Z_i}(T_i)\right)^2\right]\right] \leq \frac{1}{4}\left(\frac{1}{|X_i|} + \frac{1}{|Z_i|}\right).$$

Recognizing that $L_i$ is non-negative, we conclude

$$0 \leq \mathbb{E}\left[L_i\left(\{I\}_{i=1}^m\right)\right] \leq \frac{1}{4}\left(\frac{1}{|X_i|} + \frac{1}{|Z_i|}\right).$$

For the second part we assume that $\Pi \in \mathcal{M}_1(\mathcal{M}_1^c(\mathbb{R}))$, i.e. $\Pi$ is a distribution over the set of continuous $\mathbb{R}$-valued probability distributions. Again conditioning on $\mathcal{P}$, we can now apply the second part of Lemma 2 to the inner expectation to get

$$\mathbb{E}_{\mathcal{P}}\left[\mathbb{E}_{X_i, Z_i, T_i}\left[\left(F_{X_i}(T_i) - F_{Z_i}(T_i)\right)^2\right]\right] = \frac{1}{6}\left(\frac{1}{|X_i|} + \frac{1}{|Z_i|}\right)$$

so the upper bound improves to

$$\mathbb{E}\left[L_i\left(\{I\}_{i=1}^m\right)\right] \leq \frac{1}{6}\left(\frac{1}{|X_i|} + \frac{1}{|Z_i|}\right).$$

For the lower bound notice that $\sigma(X_{i,1}, \ldots, X_{i,n_i}, T_i) \subseteq \sigma(X_{i,1}, \ldots, X_{i,n_i}, T_i, \mathcal{P})$. Therefore Lemma 5 tells us that

$$\mathbb{E}\left[L_i\left(\{I\}_{i=1}^m\right)\right] = \mathbb{E}\left[\left(\mathbb{E}\left[F_{Z_i}(T_i)\,|\,X_{i,1}, \ldots, X_{i,n_i}, T_i\right] - F_{Z_i}(T_i)\right)^2\right]$$

$$\geq \mathbb{E}\left[\left(\mathbb{E}\left[F_{Z_i}(T_i)\,|\,X_{i,1}, \ldots, X_{i,n_i}, T_i, \mathcal{P}\right] - F_{Z_i}(T_i)\right)^2\right].$$

But appealing to Lemmas 3 then 1 (using that $\mathcal{P} \in \mathcal{M}_1^c(\mathbb{R})$) gives

$$\mathbb{E}\left[\left(\mathbb{E}\left[F_{Z_i}(T_i)\,|\,X_{i,1}, \ldots, X_{i,n_i}, T_i, \mathcal{P}\right] - F_{Z_i}(T_i)\right)^2\right] = \mathbb{E}\left[\left(F_{\mathcal{P}}(T_i) - F_{Z_i}(T_i)\right)^2\right]$$

$$= \mathbb{E}_{\mathcal{P}}\left[\mathbb{E}_{Z_i, T_i}\left[\left(F_{\mathcal{P}}(T_i) - F_{Z_i}(T_i)\right)^2\right]\right]$$

$$= \mathbb{E}_{\mathcal{P}}\left[\frac{1}{6\,|Z_i|}\right]$$

$$= \frac{1}{6\,|Z_i|}$$

which concludes the proof of the lower bound. $\qquad\square$

## C.2 Proofs omitted from Subsection 2.2

**Theorem 2.** *The mechanism in Algorithm 2 satisfies truthfulness. Moreover, if there is a feature $k \in [K]$, for which $\Pi$ is not degenerate, then Algorithm 2 also satisfies MIB.*

*Proof.* For truthfulness we refer to Proposition 7.

For $n = (n_1, \ldots, n_m) \in \mathbb{N}^m$, let $L_i\left(n, \{I\}_{i=1}^m\right)$ denote the value of $L_i$ in Algorithm 2 when agent $j \in [m]$ has $n_j$ data points and agents use $\{I\}_{i=1}^m \in \mathcal{F}^m$. Proposition 8 tells us that

$$\mathbb{E}\left[L_i\left(n, \{I\}_{i=1}^m\right)\right] - \mathbb{E}\left[L_i\left(n + e_i, \{I\}_{i=1}^m\right)\right]$$

$$= \frac{1}{K}\sum_{k=1}^K \mathbb{E}\left[\left(\mathbb{E}\left[F_{Z_i^k}(T_i^k)\,|\,\mathcal{G}_{n_i}^k\right] - \mathbb{E}\left[F_{Z_i^k}(T_i^k)\,|\,\mathcal{G}_{n_i+1}^k\right]\right)^2\right].$$

where $\mathcal{G}_j^k = \sigma\left(X_{i,1},\ldots X_{i,j}, T_i^k\right)$. Also observe that

$$P\left(Z_{i,1}^k \leq T_i^k | X_{i,1},\ldots,X_{i,n_i}, T_i^k\right) = \mathbb{E}\left[F_{Z_i^k}\left(T_i^k\right)|\mathcal{G}_{n_i}^k\right],$$

$$P\left(Z_{i,1}^k \leq T_i^k | X_{i,1},\ldots,X_{i,n_i+1}, T_i^k\right) = \mathbb{E}\left[F_{Z_i^k}\left(T_i^k\right)|\mathcal{G}_{n_i+1}^k\right].$$

Since we assume there is a feature $k \in [K]$ for which $\Pi$ is non-degenerate, the conditional probabilities are not almost surely equal for at least one feature. Therefore,

$$\mathbb{E}\left[L_i\left(n, \{I\}_{i=1}^m\right)\right] - \mathbb{E}\left[L_i\left(n + e_i, \{I\}_{i=1}^m\right)\right] > 0$$

so the MIB property is satisfied.

$\square$

**Proposition 7.** *Let $L_i\left(\{f_i\}_{i=1}^m\right)$ denote the value of $L_i$ in Algorithm 2 when all agents use $\{f_i\}_{i=1}^m \in \mathcal{F}^m$. Then, for any $f_i \in \mathcal{F}$, $\mathbb{E}\left[L_i\left(\{I\}_{i=1}^m\right)\right] \leq \mathbb{E}\left[L_i\left(f_i, \{I\}_{j\neq i}\right)\right]$.*

*Proof.* By the definition of Algorithm 2 we have

$$\mathbb{E}\left[L_i\left(f_i, \{I\}_{j\neq i}\right)\right] = \frac{1}{K}\sum_{k=1}^K \mathbb{E}\left[L_i^k\left(f_i, \{I\}_{j\neq i}\right)\right].$$

By definition $\mathbb{E}\left[F_{Z_i^k}\left(T_i^k\right)|X_{i,1},\ldots,X_{i,|Y_i|}, T_i^k\right]$ is $\left(X_{i,1},\ldots,X_{i,|Y_i|}, T_i^k\right)$-measurable, so there exists a measurable function $g : \mathcal{X}^{|Y_i|} \times \mathbb{R} \to \mathbb{R}$ such that

$$g\left(X_{i,1},\ldots,X_{i,|Y_i|}, T_i^k\right) = \mathbb{E}\left[F_{Z_i^k}\left(T_i^k\right)|X_{i,1},\ldots,X_{i,|Y_i|}, T_i^k\right].$$

The conditional expectation

$$\mathbb{E}\left[F_{Z_i^k}\left(T_i^k\right)|X_{i,1} = Y_{i,1},\ldots,X_{i,|Y_i|} = Y_{i,|Y_i|}, T_i^k\right]$$

is shorthand for $g\left(Y_{i,1},\ldots,Y_{i,|Y_i|}, T_i^k\right)$. Since we assume $f_i$ is measurable,

$$\left(Y_{i,1},\ldots,Y_{i,|Y_i|}\right) = f_i\left(X_{i,1},\ldots,X_{i,n_i}\right)$$

is $\left(X_{i,1},\ldots,X_{i,n_i}\right)$-measurable. Therefore, we know that

$$g\left(Y_{i,1},\ldots,Y_{i,|Y_i|}, T_i^k\right) = \mathbb{E}\left[F_{Z_i^k}\left(T_i^k\right)|X_{i,1} = Y_{i,1},\ldots,X_{i,|Y_i|} = Y_{i,|Y_i|}, T_i^k\right]$$

is $\left(X_{i,1},\ldots,X_{i,n_i}, T_i^k\right)$-measurable. This lets us apply Lemma 5 to get

$$
\begin{aligned}
&\mathbb{E}\left[L_i^k\left(f_i, \{I\}_{j\neq i}\right)\right] \\
&= \mathbb{E}\left[\left(\mathbb{E}\left[F_{Z_i^k}\left(T_i^k\right)|X_{i,1} = Y_{i,1},\ldots,X_{i,|Y_i|} = Y_{i,|Y_i|}, T_i^k\right] - F_{Z_i^k}\left(T_i^k\right)\right)^2\right] \\
&\geq \mathbb{E}\left[\left(\mathbb{E}\left[F_{Z_i^k}\left(T_i^k\right)|X_{i,1},\ldots,X_{i,n_i}, T_i^k\right] - F_{Z_i^k}\left(T_i^k\right)\right)^2\right] \\
&= \mathbb{E}\left[L_i^k\left(\{I\}_{i=1}^m\right)\right].
\end{aligned}
$$

Repeatedly applying this argument for each feature gives us

$$\frac{1}{K}\sum_{k=1}^K \mathbb{E}\left[L_i^k\left(f_i, \{I\}_{j\neq i}\right)\right] \geq \frac{1}{K}\sum_{k=1}^K \mathbb{E}\left[L_i^k\left(\{I\}_{i=1}^m\right)\right] = \mathbb{E}\left[L_i\left(\{I\}_{i=1}^m\right)\right].$$

$\square$

**Proposition 8.** *For $n = (n_1, \ldots, n_m) \in \mathbb{N}^m$ let $L_i\left(n, \{I\}_{i=1}^m\right)$ denote the value of $L_i$ in Algorithm 2 when agent $j \in [m]$ has $n_j$ data points and agents use $\{I\}_{i=1}^m \in \mathcal{F}^m$. Then*

$$\mathbb{E}\left[L_i\left(n, \{I\}_{i=1}^m\right)\right] - \mathbb{E}\left[L_i\left(n + e_i, \{I\}_{i=1}^m\right)\right]$$

$$= \frac{1}{K} \sum_{k=1}^K \mathbb{E}\left[\left(\mathbb{E}\left[F_{Z_i^k}\left(T_i^k\right) | \mathcal{G}_{n_i}^k\right] - \mathbb{E}\left[F_{Z_i^k}\left(T_i^k\right) | \mathcal{G}_{n_i+1}^k\right]\right)^2\right]$$

*where $\mathcal{G}_j^k = \sigma\left(X_{i,1}, \ldots X_{i,j}, T_i^k\right)$.*

*Proof.* By the definition of Algorithm 2

$$\mathbb{E}\left[L_i\left(n, \{I\}_{i=1}^m\right)\right] = \frac{1}{K} \sum_{k=1}^K \mathbb{E}\left[L_i^k\left(n, \{I\}_{i=1}^m\right)\right]$$

$$= \frac{1}{K} \sum_{k=1}^K \mathbb{E}\left[\left(\mathbb{E}\left[F_{Z_i^k}\left(T_i^k\right) | \mathcal{G}_{n_i}^k\right] - F_{Z_i^k}\left(T_i^k\right)\right)^2\right]$$

Let $U^k = F_{Z_i^k}\left(T_i^k\right)$. From the equation above, the tower property and definition of conditional variance tell us that

$$\mathbb{E}\left[L_i\left(n, \{I\}_{i=1}^m\right)\right] = \frac{1}{K} \sum_{k=1}^K \mathbb{E}\left[\mathrm{Var}\left(U^k | \mathcal{G}_{n_i}^k\right)\right].$$

An analogous argument gives

$$\mathbb{E}\left[L_i\left(n + e_i, \{I\}_{i=1}^m\right)\right] = \frac{1}{K} \sum_{k=1}^K \mathbb{E}\left[\mathrm{Var}\left(U^k | \mathcal{G}_{n_i+1}^k\right)\right].$$

Let $Y^k = \mathbb{E}\left[U^k | \mathcal{G}_{n_i+1}^k\right]$. We can now repeatedly appeal to Lemma 4 to get

$$\mathbb{E}\left[L_i\left(n, \{I\}_{i=1}^m\right)\right] - \mathbb{E}\left[L_i\left(n + e_i, \{I\}_{i=1}^m\right)\right]$$

$$= \frac{1}{K} \sum_{k=1}^K \left(\mathbb{E}\left[\mathrm{Var}\left(U^k | \mathcal{G}_{n_i}^k\right)\right] - \mathbb{E}\left[\mathrm{Var}\left(U^k | \mathcal{G}_{n_i+1}^k\right)\right]\right)$$

$$= \frac{1}{K} \sum_{k=1}^K \mathbb{E}\left[\mathrm{Var}\left(Y^k | \mathcal{G}_{n_i}^k\right)\right].$$

Using the tower property now lets us conclude that

$$\frac{1}{K} \sum_{k=1}^K \mathbb{E}\left[\mathrm{Var}\left(Y^k | \mathcal{G}_{n_i}^k\right)\right] = \frac{1}{K} \sum_{k=1}^K \mathbb{E}\left[\mathbb{E}\left[\left(Y^k - \mathbb{E}\left[Y^k | \mathcal{G}_{n_i}^k\right]\right)^2 | \mathcal{G}_{n_i}^k\right]\right]$$

$$= \frac{1}{K} \sum_{k=1}^K \mathbb{E}\left[\left(\mathbb{E}\left[U^k | \mathcal{G}_{n_i}^k\right] - \mathbb{E}\left[U^k | \mathcal{G}_{n_i+1}^k\right]\right)^2\right]$$

$$= \frac{1}{K} \sum_{k=1}^K \mathbb{E}\left[\left(\mathbb{E}\left[F_{Z_i^k}\left(T_i^k\right) | \mathcal{G}_{n_i}^k\right] - \mathbb{E}\left[F_{Z_i^k}\left(T_i^k\right) | \mathcal{G}_{n_i+1}^k\right]\right)^2\right].$$

$\square$

**Proposition 9.** *Let $L_i\left(\{I\}_{i=1}^m\right)$ denote the value of $L_i$ when all agents are truthful in Algorithm 2. Then, $0 \leq \mathbb{E}\left[L_i\left(\{I\}_{i=1}^m\right)\right] \leq \frac{1}{4}\left(\frac{1}{|X_i|} + \frac{1}{|Z_i|}\right)$. Moreover, if $\forall k \in [K]$, $\mathcal{P}^k = \mathcal{P} \circ \left(\varphi^k\right)^{-1}$ is a.s. continuous, then $\frac{1}{6|Z_i|} \leq \mathbb{E}\left[L_i\left(\{I\}_{i=1}^m\right)\right] \leq \frac{1}{6}\left(\frac{1}{|X_i|} + \frac{1}{|Z_i|}\right)$.*

*Proof.* By definition

$$\mathbb{E}\left[L_i\left(\{I\}_{i=1}^m\right)\right] = \frac{1}{K}\sum_{k=1}^K \mathbb{E}\left[L_i^k\left(\{I\}_{i=1}^m\right)\right].$$

Define $X_i^k = \left(\varphi^k\left(X_{i,j}\right)\right)_{j=1}^{n_i}$. We have $F_{X_i^k}\left(T_i^k\right)$ is $\left(X_{i,1},\ldots,X_{i,n_i},T_i^k\right)$-measurable. Therefore, Lemma 5 tells us that

$$\mathbb{E}\left[L_i^k\left(\{I\}_{i=1}^m\right)\right] = \mathbb{E}\left[\left(\mathbb{E}\left[F_{Z_i^k}\left(T_i^k\right)\mid X_{i,1},\ldots,X_{i,n_i},T_i^k\right] - F_{Z_i^k}\left(T_i^k\right)\right)^2\right]$$

$$\leq \mathbb{E}\left[\left(F_{X_i^k}\left(T_i^k\right) - F_{Z_i^k}\left(T_i^k\right)\right)^2\right]$$

$$= \mathbb{E}_{\mathcal{P}}\left[\mathbb{E}_{X_i^k,Z_i^k,T_i^k}\left[\left(F_{X_i^k}\left(T_i^k\right) - F_{Z_i^k}\left(T_i^k\right)\right)^2\right]\right].$$

Applying the first part of Lemma 2 to the inner expectation gives

$$\mathbb{E}_{\mathcal{P}}\left[\mathbb{E}_{X_i^k,Z_i^k,T_i^k}\left[\left(F_{X_i^k}\left(T_i^k\right) - F_{Z_i^k}\left(T_i^k\right)\right)^2\right]\right] \leq \frac{1}{4}\left(\frac{1}{|X_i^k|} + \frac{1}{|Z_i^k|}\right) = \frac{1}{4}\left(\frac{1}{|X_i|} + \frac{1}{|Z_i|}\right)$$

so we conclude

$$0 \leq \mathbb{E}\left[L_i\left(\{I\}_{i=1}^m\right)\right] \leq \frac{1}{4}\left(\frac{1}{|X_i|} + \frac{1}{|Z_i|}\right).$$

For the second part, when we assume $\mathcal{P}^k$ is a.s. continuous, we can apply the second part of Lemma 2 to get

$$\mathbb{E}_{\mathcal{P}}\left[\mathbb{E}_{X_i^k,Z_i^k,T_i^k}\left[\left(F_{X_i^k}\left(T_i^k\right) - F_{Z_i^k}\left(T_i^k\right)\right)^2\right]\right] = \frac{1}{6}\left(\frac{1}{|X_i^k|} + \frac{1}{|Z_i^k|}\right) = \frac{1}{6}\left(\frac{1}{|X_i|} + \frac{1}{|Z_i|}\right)$$

so the upper bound improves to

$$\mathbb{E}\left[L_i\left(\{I\}_{i=1}^m\right)\right] \leq \frac{1}{6}\left(\frac{1}{|X_i|} + \frac{1}{|Z_i|}\right).$$

For the lower bound, note that $\sigma\left(X_{i,1},\ldots,X_{i,n_i},T_i^k\right) \subseteq \sigma\left(X_{i,1},\ldots,X_{i,n_i},T_i^k,\mathcal{P}^k\right)$ so Lemma 5 gives us that

$$\mathbb{E}\left[L_i^k\left(\{I\}_{i=1}^m\right)\right] = \mathbb{E}\left[\left(\mathbb{E}\left[F_{Z_i^k}\left(T_i^k\right)\mid X_{i,1},\ldots,X_{i,n_i},T_i^k\right] - F_{Z_i^k}\left(T_i^k\right)\right)^2\right]$$

$$\geq \mathbb{E}\left[\left(\mathbb{E}\left[F_{Z_i^k}\left(T_i^k\right)\mid X_{i,1},\ldots,X_{i,n_i},T_i^k,\mathcal{P}^k\right] - F_{Z_i^k}\left(T_i^k\right)\right)^2\right].$$

Now appealing to Lemmas 3 then 1 gives

$$\mathbb{E}\left[\left(\mathbb{E}\left[F_{Z_i^k}\left(T_i^k\right)\mid X_{i,1},\ldots,X_{i,n_i},T_i^k,\mathcal{P}^k\right] - F_{Z_i^k}\left(T_i^k\right)\right)^2\right]$$

$$= \mathbb{E}\left[\left(F_{\mathcal{P}^k}\left(T_i^k\right) - F_{Z_i^k}\left(T_i^k\right)\right)^2\right]$$

$$= \mathbb{E}_{\mathcal{P}^k}\left[\mathbb{E}_{Z_i^k,T_i^k}\left[\left(F_{\mathcal{P}^k}\left(T_i^k\right) - F_{Z_i^k}\left(T_i^k\right)\right)^2\right]\right]$$

$$= \mathbb{E}_{\mathcal{P}^k}\left[\frac{1}{6\left|Z_i^k\right|}\right]$$

$$= \frac{1}{6\left|Z_i\right|}.$$

Therefore, we conclude that $\frac{1}{6|Z_i|} \leq \mathbb{E}\left[L_i\left(\{I\}_{i=1}^m\right)\right].$ □

# D   Proofs omitted from Section 3

**Theorem 3.** *The mechanism in Algorithm 3 is $\frac{1}{4}\left(\frac{1}{|X_i|+|W_i|}+\frac{1}{|Z_i|}\right)$-approximately truthful in both the Bayesian and frequentist settings. Moreover, if there is a feature $k \in [K]$, for which $\Pi$ is not degenerate, then Algorithm 3 satisfies MIB in the Bayesian setting. If it is not the case that $\mathcal{C} \subseteq \left\{\mathcal{P} \in \mathcal{M}_1\left(\mathcal{X}\right) : \forall k \in [K], \mathcal{P} \circ \left(\varphi^k\right)^{-1} \in \delta_x, x \in \mathbb{R}\right\}$ then Algorithm 3 satisfies MIB in the frequentist setting.*

*Proof.* For $\frac{1}{4}\left(\frac{1}{|X_i|+|W_i|}+\frac{1}{|Z_i|}\right)$-approximate truthfulness we refer to Proposition 11.

Let $\mathcal{P}^k = \mathcal{P} \circ \left(\varphi^k\right)^{-1}$. For MIB we first look at the Bayesian setting and then the frequentist setting.

For the Bayesian setting, from the assumption about $\Pi$ we know that $\exists k \in [K]$ where $\forall n_i \in \mathbb{N}$ it is not the case that

$$P\left(Z_{i,1}^k \leq T_i^k | X_{i,1}, \ldots, X_{i,n_i}, T_i^k\right) \overset{a.s.}{=} P\left(Z_{i,1}^k \leq T_i^k | X_{i,1}, \ldots, X_{i,n_i+1}, T_i^k\right).$$

Now notice that this implies that for at least one of the $k \in [K]$ features, $P\left(\mathcal{P}^k \in \{\delta_x : x \in \mathcal{X}\}\right) < 1$, or else the conditional probabilities above would automatically be equal for each $k \in [K]$.

We know from Proposition 12 that

$$\mathbb{E}\left[L_i\left(n, \{I\}_{i=1}^m\right)\right] - \mathbb{E}\left[L_i\left(n + e_i, \{I\}_{i=1}^m\right)\right]$$

$$= \frac{1}{K}\sum_{k=1}^K \mathbb{E}\left[F_{\mathcal{P}^k}\left(T_i^k\right)\left(1 - F_{\mathcal{P}^k}\left(T_i^k\right)\right)\right]\left(\frac{1}{n_i + |W_i|} - \frac{1}{n_i + 1 + |W_i|}\right).$$

But notice that

$$\exists k \in [K] \quad s.t. \quad P\left(\mathcal{P}^k \in \{\delta_x : x \in \mathcal{X}\}\right) < 1$$

implies $\frac{1}{K}\sum_{k=1}^K \mathbb{E}\left[F_{\mathcal{P}^k}\left(T_i^k\right)\left(1 - F_{\mathcal{P}^k}\left(T_i^k\right)\right)\right] > 0$. Therefore

$$\mathbb{E}\left[L_i\left(n, \{I\}_{i=1}^m\right)\right] - \mathbb{E}\left[L_i\left(n + e_i, \{I\}_{i=1}^m\right)\right] > 0$$

which proves MIB for the Bayesian setting.

For the frequentist setting, we have from Proposition 13 that

$$\sup_{\mathcal{P} \in \mathcal{C}} \mathbb{E}\left[L_i\left(n, \{I\}_{i=1}^m\right)\right] - \sup_{\mathcal{P} \in \mathcal{C}} \mathbb{E}\left[L_i\left(n + e_i, \{I\}_{i=1}^m\right)\right]$$

$$= \left(\sup_{\mathcal{P} \in \mathcal{C}} \frac{1}{K}\sum_{k=1}^K \mathbb{E}\left[F_{\mathcal{P}^k}\left(T_i^k\right)\left(1 - F_{\mathcal{P}^k}\left(T_i^k\right)\right)\right]\right)\left(\frac{1}{n_i + |W_i|} - \frac{1}{n_i + 1 + |W_i|}\right).$$

If it is not the case that

$$\mathcal{C} \subseteq \left\{\mathcal{P} \in \mathcal{M}_1\left(\mathcal{X}\right) : \forall k \in [K], \mathcal{P} \circ \left(\varphi^k\right)^{-1} \in \delta_x, x \in \mathbb{R}\right\}$$

then

$$\sup_{\mathcal{P} \in \mathcal{C}} \frac{1}{K}\sum_{k=1}^K \mathbb{E}\left[F_{\mathcal{P}^k}\left(T_i^k\right)\left(1 - F_{\mathcal{P}^k}\left(T_i^k\right)\right)\right] > 0$$

so we find

$$\sup_{\mathcal{P} \in \mathcal{C}} \mathbb{E}\left[L_i\left(n, \{I\}_{i=1}^m\right)\right] - \sup_{\mathcal{P} \in \mathcal{C}} \mathbb{E}\left[L_i\left(n + e_i, \{I\}_{i=1}^m\right)\right] > 0$$

which proves MIB for the frequentist setting.

$\square$

**Proposition 10.** *Let $L_i\left(\{I\}_{i=1}^m\right)$ denote the value of $L_i$ when all agents follow $\{I\}_{i=1}^m \in \mathcal{F}^m$ in Algorithm 3. Then,*

$$0 \leq \mathop{\mathbb{E}}_{\mathcal{P}\sim\Pi}\left[\mathbb{E}_{\mathcal{P}}\left[L_i\left(\{I\}_{i=1}^m\right)\right]\right], \ \sup_{\mathcal{P}\in\mathcal{C}} \mathbb{E}_{\mathcal{P}}\left[L_i\left(\{I\}_{i=1}^m\right)\right] \leq \frac{1}{4}\left(\frac{1}{|X_i|+|W_i|} + \frac{1}{|Z_i|}\right).$$

*Moreover, if $\forall k \in [K]$, $\mathcal{P}^k = \mathcal{P} \circ \left(\varphi^k\right)^{-1}$ is a.s. continuous, then*

$$\mathop{\mathbb{E}}_{\mathcal{P}\sim\Pi}\left[\mathop{\mathbb{E}}_{\{X_i\}_i\sim\mathcal{P}}\left[L_i\left(\{I\}_{i=1}^m\right)\right]\right] = \sup_{\mathcal{P}\in\mathcal{C}} \mathbb{E}_{\mathcal{P}}\left[L_i\left(\{I\}_{i=1}^m\right)\right] = \frac{1}{6}\left(\frac{1}{|X_i|+|W_i|} + \frac{1}{|Z_i|}\right).$$

*Proof.* By the definition of Algorithm 3 we have

$$\mathop{\mathbb{E}}_{\mathcal{P}\sim\Pi}\left[\mathop{\mathbb{E}}_{\{X_i\}_i\sim\mathcal{P}}\left[L_i\left(\{I\}_{i=1}^m\right)\right]\right] = \mathop{\mathbb{E}}_{\mathcal{P}\sim\Pi}\left[\frac{1}{K}\sum_{k=1}^K \mathbb{E}\left[\left(F_{\left(Y_i^k, W_i^k\right)}\left(T_i^k\right) - F_{Z_i^k}\left(T_i^k\right)\right)^2\right]\right]$$

and

$$\sup_{\mathcal{P}\in\mathcal{C}} \mathbb{E}_{\mathcal{P}}\left[L_i\left(\{I\}_{i=1}^m\right)\right] = \sup_{\mathcal{P}\in\mathcal{C}} \frac{1}{K}\sum_{k=1}^K \mathbb{E}\left[\left(F_{\left(Y_i^k, W_i^k\right)}\left(T_i^k\right) - F_{Z_i^k}\left(T_i^k\right)\right)^2\right].$$

The first part of Lemma 2 tells us that in both the frequentist and Bayesian setting,

$$\mathbb{E}\left[\left(F_{\left(Y_i^k, W_i^k\right)}\left(T_i^k\right) - F_{Z_i^k}\left(T_i^k\right)\right)^2\right] \leq \frac{1}{4}\left(\frac{1}{|X_i|+|W_i|} + \frac{1}{|Z_i|}\right)$$

so we find

$$0 \leq \mathop{\mathbb{E}}_{\mathcal{P}\sim\Pi}\left[\mathbb{E}_{\mathcal{P}}\left[L_i\left(\{I\}_{i=1}^m\right)\right]\right], \ \sup_{\mathcal{P}\in\mathcal{C}} \mathbb{E}_{\mathcal{P}}\left[L_i\left(\{I\}_{i=1}^m\right)\right] \leq \frac{1}{4}\left(\frac{1}{|X_i|+|W_i|} + \frac{1}{|Z_i|}\right).$$

When $\forall k \in [K]$, $\mathcal{P}^k$ is a.s. continuous, we apply the second part of Lemma 2 to get

$$\mathbb{E}\left[\left(F_{\left(Y_i^k, W_i^k\right)}\left(T_i^k\right) - F_{Z_i^k}\left(T_i^k\right)\right)^2\right] = \frac{1}{6}\left(\frac{1}{|X_i|+|W_i|} + \frac{1}{|Z_i|}\right)$$

in both the frequentist and Bayesian setting. Under this additional hypothesis, we get

$$\mathop{\mathbb{E}}_{\mathcal{P}\sim\Pi}\left[\mathop{\mathbb{E}}_{\{X_i\}_i\sim\mathcal{P}}\left[L_i\left(\{I\}_{i=1}^m\right)\right]\right] = \sup_{\mathcal{P}\in\mathcal{C}} \mathbb{E}_{\mathcal{P}}\left[L_i\left(\{I\}_{i=1}^m\right)\right] = \frac{1}{6}\left(\frac{1}{|X_i|+|W_i|} + \frac{1}{|Z_i|}\right).$$

$\square$

**Proposition 11.** *Let $L_i\left(\{f_i\}_{i=1}^m\right)$ denote the value of $L_i$ in Algorithm 3 when agents use $\{f_i\}_{i=1}^m \in \mathcal{F}^m$. Let $\Pi$ be a Bayesian prior, and $\mathcal{C} \subseteq \mathcal{M}_1\left(\mathcal{X}\right)$ a class of $\mathcal{X}$-valued distributions. Then, for any $f_i \in \mathcal{F}$*

$$\mathop{\mathbb{E}}_{\mathcal{P}\sim\Pi}\left[\mathop{\mathbb{E}}_{\{X_i\}_i\sim\mathcal{P}}\left[L_i\left(\{I\}_{i=1}^m\right)\right]\right] \leq \mathop{\mathbb{E}}_{\mathcal{P}\sim\Pi}\left[\mathop{\mathbb{E}}_{\{X_i\}_i\sim\mathcal{P}}\left[L_i\left(f_i, \{I\}_{j\neq i}\right)\right]\right] + \varepsilon \quad and$$

$$\sup_{\mathcal{P}\in\mathcal{C}} \mathop{\mathbb{E}}_{\{X_i\}_i\sim\mathcal{P}}\left[L_i\left(\{I\}_{i=1}^m\right)\right] \leq \sup_{\mathcal{P}\in\mathcal{C}} \mathop{\mathbb{E}}_{\{X_i\}_i\sim\mathcal{P}}\left[L_i\left(f_i, \{I\}_{j\neq i}\right)\right] + \varepsilon$$

*where $\varepsilon = \frac{1}{4}\left(\frac{1}{|X_i|+|W_i|} + \frac{1}{|Z_i|}\right)$. Moreover, if $\forall k \in [K]$, $\mathcal{P}^k = \mathcal{P} \circ \left(\varphi^k\right)^{-1}$ is a.s. continuous in the Bayesian setting and $\forall \mathcal{P} \in \mathcal{C}$ in the frequentist setting, then the above inequalities hold with $\varepsilon = \frac{1}{6(|X_i|+|W_i|)}$.*

*Proof.* The first part of the claim, when $\varepsilon = \frac{1}{4}\left(\frac{1}{|X_i|+|W_i|} + \frac{1}{|Z_i|}\right)$, follows immediately from Proposition 10 and recognizing that

$$0 \leq \mathop{\mathbb{E}}_{\mathcal{P}\sim\Pi}\left[\mathop{\mathbb{E}}_{\{X_i\}_i\sim\mathcal{P}}\left[L_i\left(f_i, \{I\}_{j\neq i}\right)\right]\right], \ \sup_{\mathcal{P}\in\mathcal{C}} \mathop{\mathbb{E}}_{\{X_i\}_i\sim\mathcal{P}}\left[L_i\left(f_i, \{I\}_{j\neq i}\right)\right].$$

Now consider when $\forall k \in [K]$, $\mathcal{P}^k$ is a.s. continuous, where $\mathcal{P}$ has either been fixed in the frequentist setting or drawn in the Bayesian setting. By the defintion of Algorithm 3 we have

$$\mathop{\mathbb{E}}_{\{X_i\}_i \sim \mathcal{P}} \left[ L_i \left( f_i, \{I\}_{j \neq i} \right) \right] = \frac{1}{K} \sum_{k=1}^{K} \mathop{\mathbb{E}}_{\{X_i\}_i \sim \mathcal{P}} \left[ L_i^k \left( f_i, \{I\}_{j \neq i} \right) \right]$$

$$= \frac{1}{K} \sum_{k=1}^{K} \mathop{\mathbb{E}}_{\{X_i\}_i \sim \mathcal{P}} \left[ \left( F_{(Y_i^k, W_i^k)} \left( T_i^k \right) - F_{Z_i^k} \left( T_i^k \right) \right)^2 \right]$$

$$= \frac{1}{K} \sum_{k=1}^{K} \mathop{\mathbb{E}}_{\{X_i^k\}_i \sim \mathcal{P}^k} \left[ \left( F_{(Y_i^k, W_i^k)} \left( T_i^k \right) - F_{Z_i^k} \left( T_i^k \right) \right)^2 \right].$$

Thus in the Bayesian setting we have

$$\mathop{\mathbb{E}}_{\mathcal{P} \sim \Pi} \left[ \mathop{\mathbb{E}}_{\{X_i\}_i \sim \mathcal{P}} \left[ L_i \left( f_i, \{I\}_{j \neq i} \right) \right] \right]$$

$$= \frac{1}{K} \sum_{k=1}^{K} \mathop{\mathbb{E}}_{\mathcal{P} \sim \Pi} \left[ \mathop{\mathbb{E}}_{\{X_i^k\}_i \sim \mathcal{P}^k} \left[ \left( F_{(Y_i^k, W_i^k)} \left( T_i^k \right) - F_{Z_i^k} \left( T_i^k \right) \right)^2 \right] \right].$$

To get a lower bound we apply Lemma 5 followed Lemmas 3 then 2 which give

$$\frac{1}{K} \sum_{k=1}^{K} \mathop{\mathbb{E}}_{\mathcal{P} \sim \Pi} \left[ \mathop{\mathbb{E}}_{\{X_i^k\}_i \sim \mathcal{P}^k} \left[ \left( F_{(Y_i^k, W_i^k)} \left( T_i^k \right) - F_{Z_i^k} \left( T_i^k \right) \right)^2 \right] \right]$$

$$\geq \frac{1}{K} \sum_{k=1}^{K} \mathop{\mathbb{E}}_{\mathcal{P} \sim \Pi} \left[ \mathop{\mathbb{E}}_{\{X_i^k\}_i \sim \mathcal{P}^k} \left[ \left( \mathbb{E} \left[ F_{(Y_i^k, W_i^k)} \left( T_i^k \right) | X_i, W_i, T_i^k, \mathcal{P}^k \right] - F_{Z_i^k} \left( T_i^k \right) \right)^2 \right] \right]$$

$$= \frac{1}{K} \sum_{k=1}^{K} \mathop{\mathbb{E}}_{\mathcal{P} \sim \Pi} \left[ \mathop{\mathbb{E}}_{\{X_i^k\}_i \sim \mathcal{P}^k} \left[ \left( F_{\mathcal{P}^k} \left( T_i^k \right) - F_{Z_i^k} \left( T_i^k \right) \right)^2 \right] \right]$$

$$= \frac{1}{K} \sum_{k=1}^{K} \mathop{\mathbb{E}}_{\mathcal{P} \sim \Pi} \left[ \frac{1}{6 \left| Z_i^k \right|} \right]$$

$$= \frac{1}{6 \left| Z_i \right|}.$$

In the frequentist setting, independence and Lemma 1 give us

$$\mathop{\mathbb{E}}_{\{X_i^k\}_i \sim \mathcal{P}^k} \left[ \left( F_{(Y_i^k, W_i^k)} \left( T_i^k \right) - F_{Z_i^k} \left( T_i^k \right) \right)^2 \right]$$

$$= \mathop{\mathbb{E}}_{\{X_i^k\}_i \sim \mathcal{P}^k} \left[ \left( F_{(Y_i^k, W_i^k)} \left( T_i^k \right) - F_{\mathcal{P}^k} \left( T_i^k \right) \right)^2 \right] + \mathop{\mathbb{E}}_{\{X_i^k\}_i \sim \mathcal{P}^k} \left[ \left( F_{Z_i^k} \left( T_i^k \right) - F_{\mathcal{P}^k} \left( T_i^k \right) \right)^2 \right]$$

$$= \mathop{\mathbb{E}}_{\{X_i^k\}_i \sim \mathcal{P}^k} \left[ \left( F_{(Y_i^k, W_i^k)} \left( T_i^k \right) - F_{\mathcal{P}^k} \left( T_i^k \right) \right)^2 \right] + \frac{1}{6 \left| Z_i^k \right|}$$

$$\geq \frac{1}{6 \left| Z_i \right|}.$$

Therefore,

$$\frac{1}{6 \left| Z_i \right|} \leq \sup_{\mathcal{P} \in \mathcal{C}} \mathop{\mathbb{E}}_{\{X_i\}_i \sim \mathcal{P}} \left[ L_i \left( f_i, \{I\}_{j \neq i} \right) \right].$$

Together we have the following lower bound in both the frequentist and Bayesian setting

$$\frac{1}{6 \left| Z_i \right|} \leq \mathop{\mathbb{E}}_{\mathcal{P} \sim \Pi} \left[ \mathop{\mathbb{E}}_{\{X_i\}_i \sim \mathcal{P}} \left[ L_i \left( f_i, \{I\}_{j \neq i} \right) \right] \right], \quad \sup_{\mathcal{P} \in \mathcal{C}} \mathop{\mathbb{E}}_{\{X_i\}_i \sim \mathcal{P}} \left[ L_i \left( f_i, \{I\}_{j \neq i} \right) \right].$$

From part two of Lemma 2 we have that when agents submit truthfully

$$\mathop{\mathbb{E}}_{\{X_i\}_i \sim \mathcal{P}} \left[ L_i \left( \{I\}_{i=1}^m \right) \right] = \frac{1}{K} \sum_{k=1}^K \mathop{\mathbb{E}}_{\{X_i\}_i \sim \mathcal{P}} \left[ L_i^k \left( \{I\}_{i=1}^m \right) \right]$$

$$= \frac{1}{K} \sum_{k=1}^K \mathop{\mathbb{E}}_{\{X_i\}_i \sim \mathcal{P}} \left[ \left( F_{\left( Y_i^k, W_i^k \right)} \left( T_i^k \right) - F_{Z_i^k} \left( T_i^k \right) \right)^2 \right]$$

$$= \frac{1}{K} \sum_{k=1}^K \frac{1}{6} \left( \frac{1}{\left| Y_i^k \right| + \left| W_i^k \right|} + \frac{1}{\left| Z_i^k \right|} \right)$$

$$= \frac{1}{6} \left( \frac{1}{|X_i| + |W_i|} + \frac{1}{|Z_i|} \right)$$

which implies that

$$\mathop{\mathbb{E}}_{\mathcal{P} \sim \Pi} \left[ \mathop{\mathbb{E}}_{\{X_i\}_i \sim \mathcal{P}} \left[ L_i \left( \{I\}_{i=1}^m \right) \right] \right] = \sup_{\mathcal{P} \in \mathcal{C}} \mathop{\mathbb{E}}_{\{X_i\}_i \sim \mathcal{P}} \left[ L_i \left( \{I\}_{i=1}^m \right) \right] = \frac{1}{6} \left( \frac{1}{|X_i| + |W_i|} + \frac{1}{|Z_i|} \right).$$

Combining this with the lower bounds, we conclude

$$\mathop{\mathbb{E}}_{\mathcal{P} \sim \Pi} \left[ \mathop{\mathbb{E}}_{\{X_i\}_i \sim \mathcal{P}} \left[ L_i \left( \{I\}_{i=1}^m \right) \right] \right] - \mathop{\mathbb{E}}_{\mathcal{P} \sim \Pi} \left[ \mathop{\mathbb{E}}_{\{X_i\}_i \sim \mathcal{P}} \left[ L_i \left( f_i, \{I\}_{j \neq i} \right) \right] \right] \leq \frac{1}{6 \left( |X_i| + |W_i| \right)}$$

$$\sup_{\mathcal{P} \in \mathcal{C}} \mathop{\mathbb{E}}_{\{X_i\}_i \sim \mathcal{P}} \left[ L_i \left( \{I\}_{i=1}^m \right) \right] - \sup_{\mathcal{P} \in \mathcal{C}} \mathop{\mathbb{E}}_{\{X_i\}_i \sim \mathcal{P}} \left[ L_i \left( f_i, \{I\}_{j \neq i} \right) \right] \leq \frac{1}{6 \left( |X_i| + |W_i| \right)}$$

which completes the proof. $\qquad\square$

**Proposition 12.** *For $n = (n_1, \ldots, n_m) \in \mathbb{N}^m$ let $L_i \left( n, \{f_i\}_{i=1}^m \right)$ denote the value of $L_i$ in Algorithm 3 when agent $j \in [m]$ has $n_j$ data points and agents use $\{f_i\}_{i=1}^m \in \mathcal{F}^m$. Then*

$$\mathbb{E} \left[ L_i \left( n, \{I\}_{i=1}^m \right) \right] - \mathbb{E} \left[ L_i \left( n + e_i, \{I\}_{i=1}^m \right) \right]$$

$$= \frac{1}{K} \sum_{k=1}^K \mathbb{E} \left[ F_{\mathcal{P}^k} \left( T_i^k \right) \left( 1 - F_{\mathcal{P}^k} \left( T_i^k \right) \right) \right] \left( \frac{1}{n_i + |W_i|} - \frac{1}{n_i + 1 + |W_i|} \right)$$

*where $\mathcal{P}^k = \mathcal{P} \circ \left( \varphi^k \right)^{-1}$.*

*Proof.* By the definition of Algorithm 3 we have

$$\mathbb{E} \left[ L_i \left( n, \{I\}_{i=1}^m \right) \right] = \frac{1}{K} \sum_{k=1}^K \mathbb{E} \left[ L_i^k \left( n, \{I\}_{i=1}^m \right) \right] = \frac{1}{K} \sum_{k=1}^K \mathbb{E} \left[ \left( F_{\left( X_i^k, W_i^k \right)} \left( T_i^k \right) - F_{Z_i^k} \left( T_i^k \right) \right)^2 \right].$$

Let $F_{\mathcal{P}^k}$ be the CDF for $\mathcal{P}^k$. We start by rewriting each term in the sum above as

$$\mathbb{E} \left[ \left( F_{\left( X_i^k, W_i^k \right)} \left( T_i^k \right) - F_{Z_i^k} \left( T_i^k \right) \right)^2 \right] = \mathop{\mathbb{E}}_{\mathcal{P}^k} \left[ \mathop{\mathbb{E}}_{X_i^k, W_i^k, Z_i^k, T_i^k} \left[ \left( F_{\left( X_i^k, W_i^k \right)} \left( T_i^k \right) - F_{Z_i^k} \left( T_i^k \right) \right)^2 \right] \right].$$

Following the same steps in Lemma 2 up to equation (5) gives us

$$\mathop{\mathbb{E}}_{X_i^k, W_i^k, Z_i^k, T_i^k} \left[ \left( F_{\left( X_i^k, W_i^k \right)} \left( T_i^k \right) - F_{Z_i^k} \left( T_i^k \right) \right)^2 \right]$$

$$= \mathop{\mathbb{E}}_{T_i^k} \left[ F_{\mathcal{P}^k} (T_i^k) \left( 1 - F_{\mathcal{P}^k} (T_i^k) \right) \right] \left( \frac{1}{n_i + |W_i|} + \frac{1}{|Z_i|} \right).$$

Therefore,

$$\mathbb{E} \left[ L_i \left( n, \{I\}_{i=1}^m \right) \right] = \frac{1}{K} \sum_{k=1}^K \mathbb{E} \left[ F_{\mathcal{P}^k} (T_i^k) \left( 1 - F_{\mathcal{P}^k} (T_i^k) \right) \right] \left( \frac{1}{n_i + |W_i|} + \frac{1}{|Z_i|} \right).$$

The same argument gives an analogous result for $\mathbb{E}\left[L_i\left(n + e_i, \{I\}_{i=1}^m\right)\right]$. Taking the difference we find

$$\mathbb{E}\left[L_i\left(n, \{I\}_{i=1}^m\right)\right] - \mathbb{E}\left[L_i\left(n + e_i, \{I\}_{i=1}^m\right)\right]$$

$$= \frac{1}{K}\sum_{k=1}^K \mathbb{E}\left[F_{\mathcal{P}^k}(T_i^k)\left(1 - F_{\mathcal{P}^k}(T_i^k)\right)\right]\left(\frac{1}{n_i + |W_i|} - \frac{1}{n_i + |W_i| + 1}\right).$$

$\square$

**Proposition 13.** *For $n = (n_1, \ldots, n_m) \in \mathbb{N}^m$ let $L_i\left(n, \{f_i\}_{i=1}^m\right)$ denote the value of $L_i$ in Algorithm 3 when agent $j \in [m]$ has $n_j$ data points and agents use $\{f_i\}_{i=1}^m \in \mathcal{F}^m$. Then*

$$\sup_{\mathcal{P}\in\mathcal{C}} \mathbb{E}\left[L_i\left(n, \{I\}_{i=1}^m\right)\right] - \sup_{\mathcal{P}\in\mathcal{C}} \mathbb{E}\left[L_i\left(n + e_i, \{I\}_{i=1}^m\right)\right]$$

$$= \left(\sup_{\mathcal{P}\in\mathcal{C}} \frac{1}{K}\sum_{k=1}^K \mathbb{E}\left[F_{\mathcal{P}^k}\left(T_i^k\right)\left(1 - F_{\mathcal{P}^k}\left(T_i^k\right)\right)\right]\right)\left(\frac{1}{n_i + |W_i|} - \frac{1}{n_i + 1 + |W_i|}\right)$$

*where $\mathcal{P}^k = \mathcal{P} \circ \left(\varphi^k\right)^{-1}$.*

*Proof.* By the definition of Algorithm 3 we have

$$\sup_{\mathcal{P}\in\mathcal{C}} \mathbb{E}\left[L_i\left(n, \{I\}_{i=1}^m\right)\right] = \sup_{\mathcal{P}\in\mathcal{C}} \frac{1}{K}\sum_{k=1}^K \mathbb{E}\left[L_i^k\left(n, \{I\}_{i=1}^m\right)\right]$$

$$= \sup_{\mathcal{P}\in\mathcal{C}} \frac{1}{K}\sum_{k=1}^K \mathbb{E}\left[\left(F_{\left(X_i^k, W_i^k\right)}\left(T_i^k\right) - F_{Z_i^k}\left(T_i^k\right)\right)^2\right].$$

Let $F_{\mathcal{P}^k}$ be the CDF for $\mathcal{P}^k$. Following the same steps in Lemma 2 up to equation (5) gives us

$$\mathbb{E}\left[\left(F_{\left(X_i^k, W_i^k\right)}\left(T_i^k\right) - F_{Z_i^k}\left(T_i^k\right)\right)^2\right] = \mathbb{E}_{T_i^k}\left[F_{\mathcal{P}^k}(T_i^k)\left(1 - F_{\mathcal{P}^k}(T_i^k)\right)\right]\left(\frac{1}{n_i + |W_i|} + \frac{1}{|Z_i|}\right).$$

Therefore,

$$\sup_{\mathcal{P}\in\mathcal{C}} \mathbb{E}\left[L_i\left(n, \{I\}_{i=1}^m\right)\right] = \sup_{\mathcal{P}\in\mathcal{C}} \frac{1}{K}\sum_{k=1}^K \mathbb{E}\left[F_{\mathcal{P}^k}(T_i^k)\left(1 - F_{\mathcal{P}^k}(T_i^k)\right)\right]\left(\frac{1}{n_i + |W_i|} + \frac{1}{|Z_i|}\right).$$

The same argument gives an analogous result for $\mathbb{E}\left[L_i\left(n + e_i, \{I\}_{i=1}^m\right)\right]$. Applying this to each feature, we find

$$\mathbb{E}\left[L_i\left(n, \{I\}_{i=1}^m\right)\right] - \mathbb{E}\left[L_i\left(n + e_i, \{I\}_{i=1}^m\right)\right]$$

$$= \sup_{\mathcal{P}\in\mathcal{C}} \frac{1}{K}\sum_{k=1}^K \mathbb{E}\left[F_{\mathcal{P}^k}(T_i^k)\left(1 - F_{\mathcal{P}^k}(T_i^k)\right)\right]\left(\frac{1}{n_i + |W_i|} + \frac{1}{|Z_i|}\right)$$

$$- \sup_{\mathcal{P}\in\mathcal{C}} \frac{1}{K}\sum_{k=1}^K \mathbb{E}\left[F_{\mathcal{P}^k}(T_i^k)\left(1 - F_{\mathcal{P}^k}(T_i^k)\right)\right]\left(\frac{1}{n_i + |W_i| + 1} + \frac{1}{|Z_i|}\right)$$

$$= \left(\sup_{\mathcal{P}\in\mathcal{C}} \frac{1}{K}\sum_{k=1}^K \mathbb{E}\left[F_{\mathcal{P}^k}\left(T_i^k\right)\left(1 - F_{\mathcal{P}^k}\left(T_i^k\right)\right)\right]\right)\left(\frac{1}{n_i + |W_i|} - \frac{1}{n_i + 1 + |W_i|}\right).$$

$\square$

# E  Examples of the conditional expectation in Algorithm 1

## E.1  The normal-normal model

**Proposition 14.** *Suppose that $\{f_i\}_{j \neq i} = \{I\}_{j \neq i}$ in Algorithm 1. Let $\mu \sim \mathcal{N}\left(a, b^2\right)$ and $X_i = \{X_{i,1}, \ldots, X_{i,n_i}\}$, $Z_i = \left\{Z_{i,1}, \ldots, Z_{i,|Z_i|}\right\}$, where $X_{i,j}, T, Z_{i,j}|p \overset{i.i.d.}{\sim} \mathcal{N}\left(\mu, \sigma^2\right)$, then*

$$\mathbb{E}\left[F_{Z_i}(T) \mid X_{i,1}, \ldots, X_{i,n_i}, T\right] = \Phi\left(\frac{T - \tilde{\mu}}{\sqrt{\sigma^2 + \tilde{\sigma}^2}}\right)$$

*where*

$$\tilde{\mu} = \frac{\frac{a}{b^2} + \frac{sum\left(X_{i,1},\ldots,X_{i,n_i},T\right)}{\sigma^2}}{\frac{1}{b^2} + \frac{n_i+1}{\sigma^2}} \quad and \quad \tilde{\sigma}^2 = \left(\frac{1}{b^2} + \frac{(n_i+1)}{\sigma^2}\right)^{-1}.$$

*Proof.* Start by noticing that the conditional expectation can be rewritten as

$$\mathbb{E}\left[F_{Z_i}(T) \mid X_{i,1},\ldots,X_{i,n_i},T\right] = \mathbb{E}\left[\frac{1}{|Z_i|}\sum_{z\in Z_i} 1_{\{z\leq T\}} \mid X_{i,1},\ldots,X_{i,n_i},T\right]$$
$$= \mathbb{E}\left[1_{\{Z_{i,1}\leq T\}} \mid X_{i,1},\ldots,X_{i,n_i},T\right]$$
$$= \mathbb{E}\left[\mathbb{E}\left[1_{\{Z_{i,1}\leq T\}} \mid \mu, X_{i,1},\ldots,X_{i,n_i},T\right] \mid X_{i,1},\ldots,X_{i,n_i},T\right].$$
(2)

where the last line follows from the tower property. By the definition of our model we know that

$$\mathbb{E}\left[1_{\{Z_{i,1}\leq T\}} \mid \mu, X_{i,1},\ldots,X_{i,n_i},T\right] = \Phi\left(\frac{T-\mu}{\sigma}\right).$$

Recall from standard normal-normal conjugacy arguments that

$$\mu|X_{i,1},\ldots X_{i,n_i},T \sim \mathcal{N}\left(\tilde{\mu},\tilde{\sigma}^2\right) \quad \text{where}$$

$$\tilde{\mu} = \frac{\frac{a}{b^2} + \frac{sum\left(X_{i,1},\ldots,X_{i,n_i},T\right)}{\sigma^2}}{\frac{1}{b^2} + \frac{n_i+1}{\sigma^2}} \quad and \quad \tilde{\sigma}^2 = \left(\frac{1}{b^2} + \frac{(n_i+1)}{\sigma^2}\right)^{-1}.$$

Therefore, we can write (2) as

$$\mathbb{E}\left[\Phi\left(\frac{T-\mu}{\sigma}\right) \mid X_{i,1},\ldots,X_{i,n_i},T\right] = \int_{-\infty}^{\infty} \Phi\left(\frac{T-\mu}{\sigma}\right)\phi_{\tilde{\mu},\tilde{\sigma}^2}(\mu)d\mu.$$

where $\phi_{\tilde{\mu},\tilde{\sigma}^2}$ is the PDF of a normal distribution with mean $\tilde{\mu}$ and variance $\tilde{\sigma}^2$. Recall the following Gaussian integral formula

$$\int_{-\infty}^{\infty} \Phi(\alpha - \beta x)\phi(x)dx = \Phi\left(\frac{\alpha}{\sqrt{1+\beta^2}}\right).$$

By the change of variables $x = \frac{\mu-\tilde{\mu}}{\tilde{\sigma}}$ we get $\Phi\left(\frac{T-\mu}{\sigma}\right) = \Phi\left(\frac{T-\tilde{\mu}}{\sigma} - \frac{\tilde{\sigma}}{\sigma}x\right)$, so applying the formula gives us

$$\int_{-\infty}^{\infty} \Phi\left(\frac{T-\mu}{\sigma}\right)\phi_{\tilde{\mu},\tilde{\sigma}^2}(\mu)d\mu = \Phi\left(\frac{T-\tilde{\mu}}{\sqrt{\sigma^2+\tilde{\sigma}^2}}\right).$$

Therefore,

$$\mathbb{E}\left[F_{Z_i}(T) \mid X_{i,1},\ldots,X_{i,n_i},T\right] = \Phi\left(\frac{T-\tilde{\mu}}{\sqrt{\sigma^2+\tilde{\sigma}^2}}\right).$$

$\square$

### E.2 The beta-bernoulli model

**Proposition 15.** *Suppose that* $\{f_i\}_{j\neq i} = \{I\}_{j\neq i}$ *in Algorithm* 1. *Let* $p \sim Beta\left(\alpha,\beta\right)$ *and* $X_i = \{X_{i,1},\ldots,X_{i,n_i}\}$, $Z_i = \left\{Z_{i,1},\ldots,Z_{i,|Z_i|}\right\}$, *where* $X_{i,j},T,Z_{i,j}|p \overset{i.i.d.}{\sim} Bern\left(p\right)$, *then*

$$\mathbb{E}\left[F_{Z_i}(T) \mid X_{i,1},\ldots,X_{i,n_i},T\right] = T + (1-T)\frac{\beta + (n_i+1) - sum\left(X_{i,1},\ldots,X_{i,n_i}\right)}{\alpha+\beta+(n_i+1)}.$$

*Proof.* Start by noticing that the conditional expectation can be rewritten as

$$\mathbb{E}\left[F_{Z_i}(T) \mid X_{i,1},\ldots,X_{i,n_i},T\right] = \mathbb{E}\left[\frac{1}{|Z_i|}\sum_{z\in Z_i} 1_{\{z\leq T\}} \mid X_{i,1},\ldots,X_{i,n_i},T\right]$$
$$= \mathbb{E}\left[1_{\{Z_{i,1}\leq T\}} \mid X_{i,1},\ldots,X_{i,n_i},T\right]$$
$$= P\left(Z_{i,1} \leq T \mid X_{i,1},\ldots,X_{i,n_i},T\right).$$

The law of total probability tells us that

$$P\left(Z_{i,1} \le T \mid X_{i,1}, \ldots, X_{i,n_i}, T\right)$$
$$= \int P\left(Z_{i,1} \le T \mid p, X_{i,1}, \ldots, X_{i,n_i}, T\right) dP\left(p \mid X_{i,1}, \ldots, X_{i,n_i}, T\right). \tag{3}$$

We now consider two cases based on whether $T$ is 0 or 1. When $T = 1$, (3) becomes

$$P\left(Z_{i,1} \le T \mid X_{i,1}, \ldots, X_{i,n_i}, T = 1\right) = \int 1 \cdot dP\left(p \mid X_{i,1}, \ldots, X_{i,n_i}, T\right) = 1.$$

When $T = 0$, recall from standard Beta-Bernoulli conjugacy arguments that

$$p \mid X_{i,1}, \ldots, X_{i,n_i}, T$$
$$\sim \mathrm{Beta}\left(\alpha + \mathrm{sum}\left(X_{i,1}, \ldots, X_{i,n_i}, T\right), \beta + (n_i + 1) - \mathrm{sum}\left(X_{i,1}, \ldots, X_{i,n_i}, T\right)\right)$$
$$= \mathrm{Beta}\left(\underbrace{\alpha + \mathrm{sum}\left(X_{i,1}, \ldots, X_{i,n_i}\right)}_{\alpha_0 :=}, \underbrace{\beta + (n_i + 1) - \mathrm{sum}\left(X_{i,1}, \ldots, X_{i,n_i}\right)}_{\beta_0 :=}\right).$$

Also observe that when $T = 0$, $P\left(Z_{i,1} \le T \mid p, X_{i,1}, \ldots, X_{i,n_i}, T\right) = 1 - p$. Therefore, (3) becomes

$$\int P\left(Z_{i,1} \le T \mid p, X_{i,1}, \ldots, X_{i,n_i}, T\right) dP\left(p \mid X_{i,1}, \ldots, X_{i,n_i}, T\right)$$
$$= \int (1 - p) \frac{p^{1-\alpha_0}(1 - p)^{1-\beta_0}}{B(\alpha_0, \beta_0)}.$$

Recall that if $Z \sim \mathrm{Beta}\left(\alpha_0, \beta_0\right)$ then

$$\frac{\alpha_0}{\alpha_0 + \beta_0} = \mathbb{E}[Z] = \int (1 - z) \frac{z^{1-\alpha_0}(1 - z)^{1-\beta_0}}{B(\alpha_0, \beta_0)}.$$

Therefore,

$$\int (1 - p) \frac{p^{1-\alpha_0}(1 - p)^{1-\beta_0}}{B(\alpha_0, \beta_0)} = \int \frac{p^{1-\alpha_0}(1 - p)^{1-\beta_0}}{B(\alpha_0, \beta_0)} - \int p \frac{p^{1-\alpha_0}(1 - p)^{1-\beta_0}}{B(\alpha_0, \beta_0)}$$
$$= 1 - \frac{\alpha_0}{\alpha_0 + \beta_0}$$
$$= \frac{\beta_0}{\alpha_0 + \beta_0}$$

so we conclude that

$$P\left(Z_{i,1} \le T \mid X_{i,1}, \ldots, X_{i,n_i}, T = 0\right) = \frac{\beta + (n_i + 1) - \mathrm{sum}\left(X_{i,1}, \ldots, X_{i,n_i}\right)}{\alpha + \beta + (n_i + 1)}.$$

Putting both cases together gives us

$$\mathbb{E}\left[F_{Z_i}(T) \mid X_{i,1}, \ldots, X_{i,n_i}, T\right] = T + (1 - T) \frac{\beta + (n_i + 1) - \mathrm{sum}\left(X_{i,1}, \ldots, X_{i,n_i}\right)}{\alpha + \beta + (n_i + 1)}.$$

$\square$

## F    Proofs of Technical results

In this section we derive a series of technical results which aid in the main proofs.

**Lemma 1.** *Let $\mathcal{P} \in \mathcal{M}_1^c(\mathbb{R})$ be a continuous probability distribution over $\mathbb{R}$, and $X = \{X_1, \ldots, X_n\}$, where $X_i, T \overset{i.i.d.}{\sim} \mathcal{P}$. Then*

$$\mathbb{E}\left[(F_X(T) - F_{\mathcal{P}}(T))^2\right] = \frac{1}{6n}.$$

*Proof.* Notice that for a fixed $t \in \mathbb{R}$,

$$\mathbb{E}\left[F_X(t)\right] = \frac{1}{n} \sum_{i=1}^{n} \mathbb{E}\left[1_{\{X_i \le t\}}\right] = F_{\mathcal{P}}(t).$$

Using this observation and noticing that $1_{\{X_i \le T\}}|T \sim \text{Bern}\left(F_{\mathcal{P}}(T)\right)$ gives

$$\begin{aligned}
\mathbb{E}\left[\left(F_X(T) - F_{\mathcal{P}}(T)\right)^2\right] &= \mathbb{E}_T\left[\mathbb{E}_X\left[\left(F_X(T) - F_{\mathcal{P}}(T)\right)^2 |T\right]\right] \\
&= \mathbb{E}_T\left[\text{Var}\left(F_X(T)|T\right)\right] \\
&= \mathbb{E}_T\left[\frac{F_{\mathcal{P}}(T)(1 - F_{\mathcal{P}}(T))}{n}\right] \\
&= \int_{-\infty}^{\infty} \frac{F_{\mathcal{P}}(T)(1 - F_{\mathcal{P}}(T))}{n} d\mathcal{P}(T).
\end{aligned}$$

Since $\mathcal{P}$ is continuous, the probability integral transform (Lemma 6) tells us that if we set $U := F_{\mathcal{P}}(T)$ then $U \sim \text{Unif}(0, 1)$. The above equation can now be written as

$$\int_{-\infty}^{\infty} \frac{F_{\mathcal{P}}(T)(1 - F_{\mathcal{P}}(T))}{n} d\mathcal{P}(T) = \int_0^1 \frac{U(1 - U)}{n} dU = \frac{1}{6n}$$

which concludes the proof. $\qquad \square$

**Lemma 2.** *Let $\mathcal{P} \in \mathcal{M}_1(\mathbb{R})$ be a probability distribution over $\mathbb{R}$, and $X = \{X_1, \ldots, X_n\}, Y = \{Y_1, \ldots, Y_m\}$ where $X_i, Y_i, T \overset{i.i.d.}{\sim} \mathcal{P}$. Then*

$$\mathbb{E}\left[\left(F_X(T) - F_Y(T)\right)^2\right] \le \frac{1}{4}\left(\frac{1}{n} + \frac{1}{m}\right).$$

*Moreover, when $\mathcal{P} \in \mathcal{M}_1^c(\mathbb{R})$*

$$\mathbb{E}\left[\left(F_X(T) - F_Y(T)\right)^2\right] = \frac{1}{6}\left(\frac{1}{n} + \frac{1}{m}\right).$$

*Proof.* We start with proving the inequality. Let $F_{\mathcal{P}}(t)$ be the CDF of $\mathcal{P}$. Notice that for a fixed $t \in \mathbb{R}$, $\mathbb{E}\left[F_X(t)\right] = \frac{1}{n}\sum_{i=1}^{n} \mathbb{E}\left[1_{\{X_i \le t\}}\right] = F_{\mathcal{P}}(t)$. Together with independence we have

$$\begin{aligned}
&\mathbb{E}\left[\left(F_X(T) - F_Y(T)\right)^2\right] \\
&= \mathbb{E}_T\left[\mathbb{E}_{X,Y}\left[\left(F_X(T) - F_Y(T)\right)^2 |T\right]\right] \\
&= \mathbb{E}_T\left[\mathbb{E}_{X,Y}\left[\left(F_X(T) - F_{\mathcal{P}}(T) + F_{\mathcal{P}}(T) - F_Y(T)\right)^2 |T\right]\right] \\
&= \mathbb{E}_T\left[\mathbb{E}_X\left[\left(F_X(T) - F_{\mathcal{P}}(T)\right)^2 |T\right] + \mathbb{E}_Y\left[\left(F_{\mathcal{P}}(T) - F_Y(T)\right)^2 |T\right]\right] \qquad (4) \\
&= \mathbb{E}_T\left[\text{Var}\left(F_X(T)|T\right) + \text{Var}\left(F_Y(T)|T\right)\right].
\end{aligned}$$

Given $T$, $F_X(T)$ and $F_Y(T)$ are sums of i.i.d. bernoulli random variables, thus

$$\begin{aligned}
\mathbb{E}_T\left[\text{Var}\left(F_X(T)|T\right) + \text{Var}\left(F_Y(T)|T\right)\right] &= \mathbb{E}_T\left[\frac{F_{\mathcal{P}}(T)(1 - F_{\mathcal{P}}(T))}{n} + \frac{F_{\mathcal{P}}(T)(1 - F_{\mathcal{P}}(T))}{m}\right] \\
&= \mathbb{E}_T\left[F_{\mathcal{P}}(T)(1 - F_{\mathcal{P}}(T))\right]\left(\frac{1}{n} + \frac{1}{m}\right) \qquad (5) \\
&\le \frac{1}{4}\left(\frac{1}{n} + \frac{1}{m}\right).
\end{aligned}$$

since $F_{\mathcal{P}}(T) \in [0, 1]$.

For the equality, we rewrite (4) and apply Lemma 1 twice to get

$$\mathbb{E}_T\left[\mathbb{E}_X\left[(F_X(T) - F_{\mathcal{P}}(T))^2\right] + \mathbb{E}_Y\left[(F_{\mathcal{P}}(T) - F_Y(T))^2\right]\right]$$
$$= \mathbb{E}_{X,T}\left[(F_X(T) - F_{\mathcal{P}}(T))^2\right] + \mathbb{E}_{Y,T}\left[(F_{\mathcal{P}}(T) - F_Y(T))^2\right]$$
$$= \frac{1}{6}\left(\frac{1}{n} + \frac{1}{m}\right).$$

$\square$

**Lemma 3.** *Let* $\Pi \in \mathcal{M}_1(\mathcal{M}_1(\mathbb{R}))$ *be a distribution over the collection of* $\mathbb{R}$*-valued distributions. Suppose that* $\mathcal{P} \sim \Pi$ *and then* $X = \{X_1, \ldots, X_n\}, Y = \{Y_1, \ldots, Y_m\}$ *where* $X_i, Y_i, T \overset{i.i.d.}{\sim} \mathcal{P}$*. Let* $F_{\mathcal{P}}(t)$ *be the CDF of* $\mathcal{P}$*. Then,*

$$\mathbb{E}\left[F_Y(T)|X_1, \ldots, X_n, T, \mathcal{P}\right] = F_{\mathcal{P}}(T).$$

*Proof.* Using conditional independence we have

$$\mathbb{E}\left[F_Y(T)|X_1, \ldots, X_n, T, \mathcal{P}\right] = \frac{1}{m}\sum_{j=1}^m \mathbb{E}\left[\mathbb{1}_{\{Y_j \leq T\}}|X_1, \ldots, X_n, T, \mathcal{P}\right]$$
$$= \frac{1}{m}\sum_{j=1}^m \mathbb{E}\left[\mathbb{1}_{\{Y_j \leq T\}}|T, \mathcal{P}\right]$$
$$= \frac{1}{m}\sum_{j=1}^m F_{\mathcal{P}}(T)$$
$$= F_{\mathcal{P}}(T).$$

$\square$

**Lemma 4.** *Let* $\mathcal{F} \subseteq \mathcal{G}$*, suppose* $X \in L^2$*, and define* $Y = \mathbb{E}[X|\mathcal{G}]$ *then*

$$\mathbb{E}\left[\text{Var}\left(X|\mathcal{F}\right)\right] - \mathbb{E}\left[\text{Var}\left(X|\mathcal{G}\right)\right] = \mathbb{E}\left[\text{Var}\left(Y|\mathcal{F}\right)\right].$$

*Proof.* Applying the law of total variation with respect to $\mathcal{F}$ and $\mathcal{G}$ gives us

$$\text{Var}(X) = \mathbb{E}\left[\text{Var}\left(X|\mathcal{G}\right)\right] + \text{Var}\left(\mathbb{E}\left[X|\mathcal{G}\right]\right) \tag{6}$$
$$\text{Var}(X) = \mathbb{E}\left[\text{Var}\left(X|\mathcal{F}\right)\right] + \text{Var}\left(\mathbb{E}\left[X|\mathcal{F}\right]\right). \tag{7}$$

Subtracting (6) from (7) gives

$$\mathbb{E}\left[\text{Var}\left(X|\mathcal{F}\right)\right] - \mathbb{E}\left[\text{Var}\left(X|\mathcal{G}\right)\right] = \text{Var}\left(\mathbb{E}\left[X|\mathcal{G}\right]\right) - \text{Var}\left(\mathbb{E}\left[X|\mathcal{F}\right]\right). \tag{8}$$

Now notice that by the tower property we have

$$\mathbb{E}\left[Y|\mathcal{F}\right] = \mathbb{E}\left[\mathbb{E}\left[X|\mathcal{G}\right]|\mathcal{F}\right] = \mathbb{E}\left[X|\mathcal{F}\right].$$

Combining this with another application of the law of total variation yields

$$\text{Var}\left(\mathbb{E}\left[X|\mathcal{G}\right]\right) = \text{Var}\left(Y\right) = \mathbb{E}\left[\text{Var}\left(Y|\mathcal{F}\right)\right] + \text{Var}\left(\mathbb{E}\left[Y|\mathcal{F}\right]\right)$$
$$= \mathbb{E}\left[\text{Var}\left(Y|\mathcal{F}\right)\right] + \text{Var}\left(\mathbb{E}\left[X|\mathcal{F}\right]\right).$$

Plugging this into the right hand side of (8) gives us

$$\mathbb{E}\left[\text{Var}\left(X|\mathcal{F}\right)\right] - \mathbb{E}\left[\text{Var}\left(X|\mathcal{G}\right)\right] = \left(\mathbb{E}\left[\text{Var}\left(Y|\mathcal{F}\right)\right] + \text{Var}\left(\mathbb{E}\left[X|\mathcal{F}\right]\right)\right) - \text{Var}\left(\mathbb{E}\left[X|\mathcal{F}\right]\right)$$
$$= \mathbb{E}\left[\text{Var}\left(Y|\mathcal{F}\right)\right].$$

$\square$

# G Known results

In this section we present two well known results and give proofs of them for completeness.

**Lemma 5** (Durrett [46] Theorem 4.1.15). *Let $X$ be a random variable such that $\mathbb{E}\left[X^2\right] < \infty$ and $\mathcal{F}$ be a $\sigma$-algebra on the underlying probability space. Then $\mathbb{E}\left[X|\mathcal{F}\right]$ is the $\mathcal{F}$-measurable random variable $Y$ which minimizes $\mathbb{E}\left[(X-Y)^2\right]$.*

*Proof.* Notice that if $Z$ is $\mathcal{F}$-measurable and $\mathbb{E}[Z^2] < \infty$ then $Z \cdot \mathbb{E}\left[X|\mathcal{F}\right] = \mathbb{E}\left[Z \cdot X|\mathcal{F}\right]$ which implies

$$\mathbb{E}\left[Z \cdot \mathbb{E}\left[X|\mathcal{F}\right]\right] = \mathbb{E}\left[\mathbb{E}\left[Z \cdot X|\mathcal{F}\right]\right] = \mathbb{E}[Z \cdot X].$$

Rearranging we find

$$\mathbb{E}\left[Z \cdot (X - \mathbb{E}[X|\mathcal{F}])\right] = 0.$$

Now suppose that $Y$ is $\mathcal{F}$-measurable and $\mathbb{E}[Y^2] < \infty$, and define $Z = \mathbb{E}[X|\mathcal{F}] - Y$. Then,

$$
\begin{aligned}
\mathbb{E}\left[(X-Y)^2\right] &= \mathbb{E}\left[(X - \mathbb{E}[X|\mathcal{F}] + Z)^2\right] \\
&= \mathbb{E}\left[(X - \mathbb{E}[X|\mathcal{F}])^2\right] + \mathbb{E}\left[Z \cdot (X - \mathbb{E}[X|\mathcal{F}])\right] + \mathbb{E}[Z^2] \\
&= \mathbb{E}\left[(X - \mathbb{E}[X|\mathcal{F}])^2\right] + \mathbb{E}[Z^2]
\end{aligned}
$$

which implies that the mean squared error is minimized when $Y = \mathbb{E}[X|\mathcal{F}]$. $\square$

**Lemma 6** (Probability integral transform). *Suppose that $X$ is a continuous $\mathbb{R}$-valued random variable. Let $U = F_X(X)$, i.e. the CDF of $X$ evaluated at $X$. Then $U \sim Unif(0,1)$.*

*Proof.* As $F_X(t)$ may not be strictly increasing, define the generalized inverse CDF $\widetilde{F}^{-1}(u) = \inf\{t \in \mathbb{R} : F_X(t) \geq u\}$. Now notice that we can write the CDF of $U$ as

$$F_U(t) = P\left(U \leq t\right) = P\left(F_X(X) \leq t\right) = P\left(X \leq \widetilde{F}^{-1}(t)\right) = F_X\left(\widetilde{F}^{-1}(t)\right) = t$$

from which we conclude that $U \sim \mathrm{Unif}(0,1)$. $\square$

