# OpenReview forum: "A Cramér–von Mises Approach to Incentivizing Truthful Data Sharing"
_NeurIPS.cc/2025/Conference — NeurIPS 2025 poster_

### Official Review · Reviewer_UVsP · 2025-06-30

**Clarity:** 4
**Significance:** 3
**Originality:** 3
**Rating:** 5
**Confidence:** 3

**Summary:**

The paper introduces a class of mechanisms to incentivize agents to truthfully share data that they have collected. In this setting, there is some underlying distribution of data, and multiple agents obtain samples of this data. The goal of the mechanism designer is to provide rewards to the agents so that they will submit as much of their true data as possible. A concern in this setting is that the agents will submit fake data (maybe simulated by a generative model) in order to obtain higher rewards. This might happen for any mechanism which simply pays more for more data. Thus, there are two key requirements in this setting. First, it must be a Nash equilibrium for the agents to submit their true data, and second, submitting more true data should yield higher rewards for each agent.

The paper provides mechanisms based on the two-sample Cramer-von Mises test that provably satisfy the above key requirements, and performs empirical evaluations based on these mechanisms. In more detail, the authors first give a mechanism where truthful data reporting is an exact Nash equilibrium. However, this mechanism relies on knowledge of a prior distribution over possible data distributions, which is unlikely to be available in many practical settings. Hence, the paper provides a second prior-free mechanism where truthful data reporting is an approximate Nash equilibrium, where the approximation error goes to zero as the size of the agents' datasets increase. Finally, the paper compares this mechanism to prior approaches in experiments with both synthetic and real data.

**Questions:**

1. The assumptions required for the class of distributions and the feature maps in the frequentist setting seem very weak. Presumably there is some quantitative dependence between how diverse the data distributions are under the feature maps, and how much more payoff an agent receives for reporting more data. Is there any such quantitative relationship that is straightforward from your current proofs?
2. Related to the above, it would be useful for the mechanism designer to be able to evaluate how good the feature maps they have chosen are. Could there be a way to empirically estimate such a dependence between diversity and marginal payoff for data?

**Ethical Concerns:**

["NO or VERY MINOR ethics concerns only"]

**Final Justification:**

The authors have answered all of my questions clearly. I maintain my opinion that the paper should be accepted.

**Limitations:**

yes

**Paper Formatting Concerns:**

None.

**Quality:**

3

**Strengths And Weaknesses:**

Strengths:
- The paper is very clear and well-written.
- The main theorems provide a practical simple mechanism to incentivize data-sharing, which nevertheless is provably correct in a very general setting (i.e. with essentially no distributional assumptions).
- As a result of the above generality, the authors show that their mechanism can be applied to many previously studied data-sharing problems, often obtaining stronger or more general results than the papers that originally introduced those problems.

Weaknesses:
- The appendix with the proofs of all the theorems is only attached in the supplementary material, so it took me a minute to find it.
- It would be nice to have quantitative assumptions on the prior/class of distributions and the feature maps, and then a quantitative conclusion on how much higher payoff an agent receives for reporting more data.

Minor typos:
- Line 257: allows them do -> allows them to
- Line 317: using on Algorithm 2 -> based on Algorithm 2

---

> ### Author Rebuttal · Authors · 2025-07-31
>
> Thank you for your response and questions.
>
> **Strengths And Weaknesses:**
>
> - *Quantifying the loss for more data:*     We do in fact have exact quantative relationships between the feature maps, data distribution, and
>     an agent's expected loss for both Algorithms 2 and 3. Because these relationships are complex, we also provide simple upper bounds for an agent's expected loss as a function of the amount
>     of data. We'll now go into more detail on this for both algorithms.
>
>      1. *For Algorithm 2*:     Proposition 9 (stated in the appendix) gives an explicit relationship for how the
>     expected loss changes when agent $i$ submits an additional data point, depending on the
>     prior and feature maps. This exactly quantifies
>     how much higher of a "payoff" an agent receives for reporting more data.
>     Proposition 10 (stated in the appendix) gives a simple upper bound for the expected loss
>     as a function of the amount of data:
>     $\frac{1}{4}
>         \left(
>             \frac{1}{
>                 |X_i|
>             }
>             +
>             \frac{1}{
>                 |Z_i|
>             }
>         \right)$
>     in general and
>     $\frac{1}{6}
>         \left(
>             \frac{1}{
>                 |X_i|
>             }
>             +
>             \frac{1}{
>                 |Z_i|
>             }
>         \right)$
>     when the selected feature maps applied to the data are continuously distributed.
>
>    2. *For Algorithm 3:* Proposition 11 (stated in the appendix) gives an analogous simple upper bound of
>     $\frac{1}{4}
>         \left(
>             \frac{1}{
>                 |X_i|+|W_i|
>             }
>             +
>             \frac{1}{
>                 |Z_i|
>             }
>         \right)$
>     for the expected loss.
>     However, when the selected feature maps applied to the data are continuously distributed,
>     this proposition provides
>     an exact expression for the expected loss in both the Bayesian and frequentist settings:
>     $\frac{1}{6}
>         \left(
>             \frac{1}{
>                 |X_i|+|W_i|
>             }
>             +
>             \frac{1}{
>                 |Z_i|
>             }
>         \right)$.
>
>    We will make sure to better highlight these quantitative relationships between the prior, feature maps,
>     amount of data, and an
>     agent's expected loss in the revision.
>
> - *Minor typos:* We will fix these in the revision.
>
> **Questions:**
>
> - *Quantitative dependence between loss, feature maps, and prior:* Yes, please see above.
>
> - *Evaluating feature maps:*  Could you please clarify what you mean by "diversity and marginal payoff for data"?

---

> > ### Author Response · Authors · 2025-08-06
> >
> > As the discussion period is almost over, we would like to know if our response has addressed your questions?

---

> ### Comment · Reviewer_UVsP · 2025-08-06
>
> Thanks for pointing out these quantitative results! I think they make the main message of the paper stronger, so it could be nice to include some mention of them in the main body.

---

### Official Review · Reviewer_jWHw · 2025-06-30

**Clarity:** 4
**Significance:** 3
**Originality:** 2
**Rating:** 4
**Confidence:** 3

**Summary:**

This paper proposes a novel class of incentive mechanisms for data sharing platforms, aimed at discouraging fabricated data submissions and rewarding truthful contributions. The core idea is to use a variant of the Cramér–von Mises (CvM) two-sample test to compare an agent's submitted data against the pooled data of other agents. The authors design both Bayesian and prior-agnostic variants, and demonstrate (approximately) truthful reporting constitutes a Nash equilibrium under mild conditions. Theoretical results are accompanied by synthetic and real-world evaluations on language and image datasets.

**Questions:**

- How is the feature map set selected in practice? In real applications with high-dimensional data (e.g., images or text), how should the designer select meaningful feature maps? How sensitive are the results to this choice?
- Can ω-approximate truthfulness be estimated empirically? Is there a practical method to assess how close the approximate Nash equilibrium is to exact truthfulness, especially in the prior-agnostic setting?
- What about privacy or decentralization? The mechanism assumes that agents' data are centrally available. Can the approach be adapted for settings where privacy or decentralization constraints exist, e.g., via secure aggregation or split computation?
- Can the mechanism handle malicious agents? What happens when a subset of agents collude or actively seek to degrade the system rather than maximize their individual rewards?
- Relation to peer-prediction and mutual information mechanisms? How does your method compare with recent information-theoretic incentive mechanisms that use mutual information, entropy, or learned models to quantify signal agreement?

**Ethical Concerns:**

["NO or VERY MINOR ethics concerns only"]

**Final Justification:**

I maintain my opinion that this paper is on the borderline; it is a sound paper but not near the top either.

**Limitations:**

Yes, the paper acknowledges its main limitations (need for feature maps, expensive Bayesian computations). However, it does not sufficiently discuss:
- Sensitivity to feature map choice
- Scalability to large agent populations
- Vulnerability to adversarial collusion
- Feasibility of real-world implementation

**Paper Formatting Concerns:**

No.

**Quality:**

4

**Strengths And Weaknesses:**

Strengths
- Sound theoretical foundation: The mechanism is rooted in well-understood statistical theory (CvM test), with careful derivation of properties like truthfulness and monotonicity in data size.
- Addressing a real gap: The paper correctly identifies a practical weakness in current incentive mechanisms that rely on restrictive assumptions or simplistic fabrication models.
- Prior-agnostic variant: The introduction of a tractable, prior-free version of the mechanism increases practical applicability, especially for complex data modalities like text or images.
- Clarity and completeness: The paper is well written, with clearly stated assumptions, formal theorems, and extensive proofs provided in the appendix.
- Empirical grounding: This work demonstrates empirical evaluations across multiple domains, including data fabricated by LLMs and diffusion models.

Weaknesses
- Limited novelty in broader context: While the use of the CvM statistic is clever, the broader idea—comparing an agent’s data to others to detect deviation—is already well established in the peer-prediction and data-market literature. The work is incremental in spirit, primarily offering a new instantiation of this idea using a specific test statistic.
- The proposed mechanisms assume that all agents' submissions can be centrally collected and compared. This ignores important privacy, communication, and scalability challenges, especially in federated or decentralized settings.
- Feature map dependency is underspecified: In high-dimensional domains (e.g., vision or language), the performance and truthfulness guarantees of the mechanism rely heavily on the choice of feature maps. While the authors allow flexibility here, no systematic guideline or analysis is offered for selecting or validating these maps.
- Approximate truthfulness is not practically quantified: The prior-agnostic mechanism achieves ω-approximate truthfulness, but ω is not empirically measured or shown to be practically negligible. This leaves a gap in understanding how robust the mechanism really is to strategic agents.
- Limited comparison with advanced baselines: The baseline mechanisms (KS, CvM, mean-diff) are simplistic. The comparison omits recent work on information-theoretic or adversarial reward mechanisms, including methods using mutual information or embedding similarity. This makes it unclear how competitive the proposed mechanism is in more realistic adversarial settings.
- Application sections are thin and idealized: The application of the mechanism to data marketplaces and federated learning is mostly conceptual and theoretical. There's no simulation of an actual marketplace, no agent utility dynamics, and no consideration of budget constraints, all of which are central to those problems.

---

> ### Author Rebuttal · Authors · 2025-07-31
>
> Thank you for your response and questions.
>
> **Strengths And Weaknesses:**
>
> - *Novelty of techniques:*     We would like to emphasize that although comparing agents' submissions is a common idea in fields
>     like peer prediction,
>     there is a substantial gap between existing methods, which are limited to simple data
>     distributions (e.g. Bernoulli, Gaussian, etc), and our approach, which supports
>     unrestricted data distributions.     In fact, we believe prior work has focused on simple distributions precisely because achieving a general result has proven to be challenging.
>     Bridging this gap requires novel insights and connections to two-sample testing
>     that are technically nontrivial.
>
> - *Privacy/communication in federated learning:* We agree that there are many practical considerations such as privacy and communication overhead that
>     go into building real world data sharing systems.
>     However, our focus is on the incentives at play and these challenges are beyond the scope of our paper.
>
> - *Choice of feature maps:*     We wish to point out, as noted in lines 219-220, that truthfulness does *not* depend on the feature maps used.
>     Our mechanism is truthful for any choice of feature maps chosen by the mechanism designer.
>     However, we allow the flexibility
>     of choosing feature maps because (i) the choice of feature maps determines how much an untruthful agent gets punished, and (ii) different types of data may be suited to different representations.
>     For low dimensional data the coordinate projections are reasonable choices for features maps,
>     whereas for high dimensional image data, embeddings given by a deep learning model are a more reasonable
>     representation.
>     The success of modern machine learning and pretrained models relies on using feature maps so our
>     approach is not atypical.
>     In our experiments, we used feature maps from existing models with no additional tweaking and found
>     our methods work well.
>
> - *Approximate truthfulness:* Theorem 3 shows that Algorithm 3 is
> $\frac{1}{4}
>         \left(
>             \frac{1}{
>                 |X_i|+|W_i|
>             }
>             +
>             \frac{1}{
>                 |Z_i|
>             }\right)$-approximately
>     truthful in both the Bayesian and frequentist settings.
>     Therefore, the more data that is being shared the more negligible $\omega$ is.
>
> - *Comparisons with advanced baselines:*     We first wish to point out that, to the best of our knowledge, we are the first theoretical
>     work on truthful data sharing to provide experiments.
>     Prior methods either do not work because their methods are not applicable for general data
>     distrbutions and/or are computationally prohibitive.
>     For instance, the method from [1] only applies to a single binary data point that is (untruthfully) reported
>     without any fabrication.
>     The approaches given by [2, 3] assume data is drawn from a Gaussian distribution and
>     [4] requires that the data distribution belongs to the Exponential family or has
>     finite support.
>     The only method that is applicable in all of our settings is the mean difference mechanism (for which theoretical guarantees are available for Gaussian problems).
>     We chose to include comparisons to KS, CvM, etc since comparisons with prior methods were not feasible.
>
>
>     Can you please clarify what you mean by "adversarial reward mechanisms"?
>
>
>     [1] Ghosh et al. (2014). Buying private data without verification.
>
>     [2] Chen et al. (2023). Mechanism Design for Collaborative Normal Mean Estimation.
>
>     [3] Clinton et al. (2025). Collaborative Mean Estimation Among Heterogeneous Strategic Agents: Individual
>     Rationality, Fairness, and Truthful Contribution.
>
>     [4] Chen et al. (2020). Truthful data acquisition via peer prediction.
>
> - *Simulation of marketplaces:* As each of these applications is its own line of work we defer in depth simulations to future work.
>
> **Questions:**
>
> - *Selecting feature maps:* Please see the "choice of feature maps" section above.
>
> - *Empirically estimating approximate truthfulness:* At the moment we do not have a way to empirically estimate how close approximate truthfulness is to
>     exact truthfulness due to the complexity of the unrestricted agent strategy space. However, we conjecture
>     that in the frequentist setting, if the class of data distributions over which the supremum is being taken
>     is sufficiently rich, then our mechanism is exactly truthful. We leave this as a question for future work.
>
> - *Privacy and decentralization:*     We imagine that techniques used to address problems of privacy and decentralization could be incorperated into
>     our mechanism but leave this as a direction for future work.
>
> - *Malicious agents:*     Data sharing with colluding agents is a direction that we are currently studying. Because our mechanism is
>     not sensitive to ourliers in data (through its use of CDFs)
>     we believe that there is an amount of robustness already present although
>     this is not quantified in the results we present.
>     We believe that techniques from robust statistics will be useful in proving results about
>     data sharing under
>     collusion although it is common in game theory not to consider adversarial agents.
>
> - *Relation to information theoretic methods:*     To the best of our knowledge, information theoretic methods that rely on concepts such as mututal information
>     suffer from either strong data distributional assumptions or are not analytically and computationally tractable. Our techniques
>     alleviate these concerns.

---

> > ### Comment · Reviewer_jWHw · 2025-08-06
> >
> > By “adversarial reward mechanisms”, it means incentive mechanisms designed to function or remain effective even when agents behave adversarially, i.e., strategically trying to game the system for personal gain rather than cooperating truthfully. In other words, those are reward systems that account for or counteract adversarial (strategic, dishonest) agent behavior in economic or game-theoretic terms.
> >
> > I recognize that addressing this issue could be challenging within the current scope of the work, especially given time constraints and practical limitations. In addition, I find the authors’ intention to leave certain aspects—such as approximation estimation in the frequentist setting, as well as considerations around privacy and decentralization—for future work to be reasonable and appropriate.

---

### Official Review · Reviewer_Ap5u · 2025-07-02

**Clarity:** 2
**Significance:** 2
**Originality:** 3
**Rating:** 4
**Confidence:** 2

**Summary:**

This paper proposes a novel incentive mechanism based on the Cramér–von Mises two-sample test to promote truthful data reporting without relying on strong distributional assumptions. The mechanism ensures (approximate) Nash equilibrium behavior and rewards larger truthful submissions. It is applicable in both Bayesian and prior-agnostic settings and is validated theoretically and empirically on synthetic, language, and image data. The approach improves robustness in data marketplaces, incentivized data collection, and federated learning.

**Questions:**

How robust is the mechanism to adversarial feature selection or manipulation of feature maps?
Can the mechanism be extended to handle collusion among agents or coordinated data fabrication?

**Ethical Concerns:**

["NO or VERY MINOR ethics concerns only"]

**Final Justification:**

Given the discussions I keep my score as it is (no major updates).

**Limitations:**

yes

**Quality:**

3

**Strengths And Weaknesses:**

Strengths:
The proposed mechanism does not rely on strong assumptions about the data distribution (e.g., Gaussian), making it applicable to a wide range of real-world scenarios.

The use of a Cramér–von Mises–inspired statistic for incentive design is novel. Theoretical guarantees (exact or approximate Nash equilibria) are rigorously established

Weaknesses:
The mechanism’s performance relies on the choice of feature maps, which must be manually selected or designed by the mechanism designer.
In the prior-agnostic setting, the mechanism only ensures approximate truthfulness.

---

> ### Author Rebuttal · Authors · 2025-07-31
>
> Thank you for your response and questions.
>
> **Strengths And Weaknesses:**
>
> - *Feature maps:* It is true that the mechanism designer will choose
>     reasonable feature maps based on the problem specification; however, this is to be expected given that there are
>     no restrictions on the data distributions supported by our mechanism.
>     Moreover, the success of modern machine learning and pretrained models relies on using feature maps so our
>     approach is not atypical.
>
> - *Approximate truthfulness:* We do have an exactly truthful mechanism in the Bayesian setting, but it can be computationally expensive (much like other methods in this space).
>     Approximate truthfulness is a trade-off for designing a prior-agnostic mechanism that is both
>     simple and tractable across diverse data distributions.
>
> **Questions:**
>
> - *Adversarial feature maps:* Because feature maps are selected by the mechanism designer we do not consider the possibility that they
>     are adversarially selected.
>     We do note however, that the that the mechanism is still truthful regardless of the feature maps selected.
>     The choice of feature maps determines how much an untruthful agent gets punished.
>
> - *Collusion among agents:*     Collusion and coordinated fabrication introduce new challenges that we are
>     investigating in future work. We believe that techniques from robust statistics can
>     be leveraged to form data sharing mechanisms that are provably robust to a small fraction of
>     strategic colluding agents.
>     Because our mechanism is
>     not sensitive to ourliers in data (through its use of CDFs)
>     we believe that there is an amount of robustness already present in our mechanism although
>     this is not a result we have provided.

---

### Official Review · Reviewer_auCC · 2025-07-03

**Clarity:** 3
**Significance:** 3
**Originality:** 3
**Rating:** 3
**Confidence:** 2

**Summary:**

The paper proposes a reward mechanism that compares each agent’s data to the aggregate of others with a statistic inspired by the two-sample Cramér–von Mises test; truthful reporting forms a Nash equilibrium in the Bayesian model. The method claims to relax Gaussian or exponential-family assumptions used in earlier peer-prediction schemes.

**Questions:**

1. Can you clarify the novelty of the method in light of existing literature?

2. Can you give precise real world guidance on implementing the method?

**Ethical Concerns:**

["NO or VERY MINOR ethics concerns only"]

**Final Justification:**

I have no further comments and I reserve the same scores.

**Limitations:**

no negative societal impact

**Quality:**

3

**Strengths And Weaknesses:**

Strength:

The problem reads quite interesting and practically relevant

The mechanism is distribution-agnostic once feature maps are chosen, extending prior work that required strong parametric forms.Initial experiments show the statistic punishes simple data-fabrication strategies on language and image corpora.

 Weakness:

The core idea already appears in peer-prediction markets and recent mean-estimation mechanisms:

https://arxiv.org/abs/2006.03992
https://arxiv.org/abs/2207.04557
https://arxiv.org/abs/1408.2539

The overall novelty can be limited in light of these existing literature.

Truthfulness still hinges on hand-picked feature maps. The paper offers no guidance or robustness study, making real world deployment unclear.

Experiments stay in toy or very small-scale settings and never involve actual data-market bargaining, so practical viability is untested.

---

> ### Author Rebuttal · Authors · 2025-07-31
>
> Thank you for your response and questions.
>
> **Strengths And Weaknesses:**
>
> - *Novelty compared to existing literature:* We have acknowledged in the paper that the idea of comparing data submitted by agents appears in prior work. However, there is a substantial gap between existing methods, which are limited to simple data distributions (e.g. Bernoulli, Gaussian, restricted class of exponential families etc), and our approach, which supports *any* data distribution. Bridging this gap requires novel insights that are technically nontrivial. In fact, we believe prior work has focused on simple distributions precisely because achieving a general result has proven to be challenging.
>
>    1. Chen et al. incentivize agents to report data
>         truthfully using information theoretic ideas from peer prediction. However, these techniques
>         require strong data distributional assumptions to gaurantee their mechanism satisfies basic properties
>         such as individual rationality or tractability.
>         Our techniques, inspired by two-sample testing, are novel (as pointed out by reviewer Ap5u) and
>         alleviate these restrictive assumptions.
>
>    2. The model studied by Karimireddy et al. differs significantly from ours because it does not allow
>         for misreporting data. The primary aim of our paper is to develop a mechanism which incentivizes agents
>         to report their data truthfully when a priori they may attempt to fabricate or misreport their data.
>
>    3. Cai et al. also do not allow for misreporting data and instead
>         adpot a model where agents can exert higher effort to decrease the noise in the data
>         they sample. They also require the
>         effort-noise tradeoff function to be known.
>         Therefore, their methods are not applicable to our problem of incentivizing agents to report truthfully
>         when dealing with data from general distributions.
>
> - *Truthfulness and feature maps:* We wish to point out, as noted in lines 219-220, that truthfulness does *not* depend on the feature maps used.
>     Our mechanism is truthful for any choice of feature maps chosen by the mechanism designer.
>     However, we allow the flexibility
>     of choosing feature maps because (i) the choice of feature maps determines how much an untruthful agent gets punished, and (ii) different types of data may be suited to different representations.
>     For low dimensional data the coordinate projections are reasonable choices for features maps,
>     whereas for high dimensional image data, embeddings given by a deep learning model are a more reasonable
>     representation.
>     The success of modern machine learning and pretrained models relies on using feature maps so our
>     approach is not atypical.
>     In our experiments, we used feature maps from existing models with no additional tweaking and found
>     our methods work well.
>
> - *Experiment settings:* As noted by Reviewer jWHw, we do provide experiments using real world image and text data.
>     While we agree that experiments involving market dynamics would be useful, we also wish to point out that,
>     to the best of our knowledge, we are the first theoretical work on truthful data sharing to provide experiments.
>     In fact, some prior methods are not even practical to implement due to expensive
>     Bayesian posterior computations, or information theoretic quantities, such as mutual information,
>     being analytically intractable for complex data distributions.
>
> **Questions:**
>
> - *Novelty compared to existing literature:* Please see above.
>
> - *Implementation guidelines:* Can you please clarify what you mean by "guidance on implementing the method"? Are you asking about how
>     to select feature maps?

---

### Decision · Program_Chairs · 2025-09-17

**Decision:**

Accept (poster)

**Comment:**

The paper introduces a novel approach for incentivizing truthful data sharing. While this topic has been studied extensively, the paper employs several innovative techniques, including a Cramer-von Mises-based loss function and the ability to handle multiple data modalities. Most of the reviewers' concerns have been successfully addressed by the authors.

What excites me about this paper, beyond the theoretical results, is the mechanism's ability to process complex data modalities.  Particularly, the authors experiment with their mechanism using text data, going far beyond previous works in this strand of research, which mostly handle simple data distributions (e.g., normal, as in Chen et al. (NeurIPS 2023)).

Adding my own judgment to the already favorable assessments of other reviewers, I recommend acceptance.